# The Illusion of Thinking:
# Understanding the Strengths and Limitations of Reasoning Models via the Lens of Problem Complexity

**Parshin Shojaee**[*][†]          **Iman Mirzadeh**[*]          **Keivan Alizadeh Vahid**

**Maxwell Horton**          **Samy Bengio**          **Mehrdad Farajtabar**

Apple

## Abstract

Recent generations of frontier language models have introduced Large Reasoning Models (LRMs) that generate detailed thinking processes before providing answers. While these models demonstrate improved performance on reasoning benchmarks, their fundamental capabilities, scaling properties, and limitations remain insufficiently understood. Current evaluations primarily focus on established mathematical and coding benchmarks, emphasizing final answer accuracy. However, this evaluation paradigm often suffers from data contamination and does not provide insights into the reasoning traces' structure and quality. In this work, we systematically investigate these gaps with the help of controllable puzzle environments that allow precise manipulation of compositional complexity while maintaining consistent logical structures. This setup enables the analysis of not only final answers but also the internal reasoning traces, offering insights into how LRMs "think". Through extensive experimentation across diverse puzzles, we show that frontier LRMs face a complete accuracy collapse beyond certain complexities. Moreover, they exhibit a counterintuitive scaling limit: their reasoning effort increases with problem complexity up to a point, then declines despite having an adequate token budget. By comparing LRMs with their standard LLM counterparts under equivalent inference compute, we identify three performance regimes: (1) low-complexity tasks where standard models surprisingly outperform LRMs, (2) medium-complexity tasks where additional thinking in LRMs demonstrates advantage, and (3) high-complexity tasks where both models experience complete collapse. We found that LRMs have limitations in exact computation: they fail to use explicit algorithms and reason inconsistently across scales and problems. We also investigate the reasoning traces in more depth, studying the patterns of explored solutions and analyzing the models' computational behavior, shedding light on their strengths, limitations, and ultimately raising questions about the nature for their reasoning capabilities.

## 1  Introduction

Large Language Models (LLMs) have recently evolved to include specialized variants explicitly designed for reasoning tasks—Large Reasoning Models (LRMs) such as OpenAI's o1/o3 [1, 2], DeepSeek-R1 [3], Claude Sonnet Thinking [4], and Gemini Thinking [5]. These models are new artifacts, characterized by their "*thinking*" mechanisms such as long Chain-of-Thought (CoT) with

---

[*]Equal contribution.

[†]Work done during an internship at Apple.

39th Conference on Neural Information Processing Systems (NeurIPS 2025).

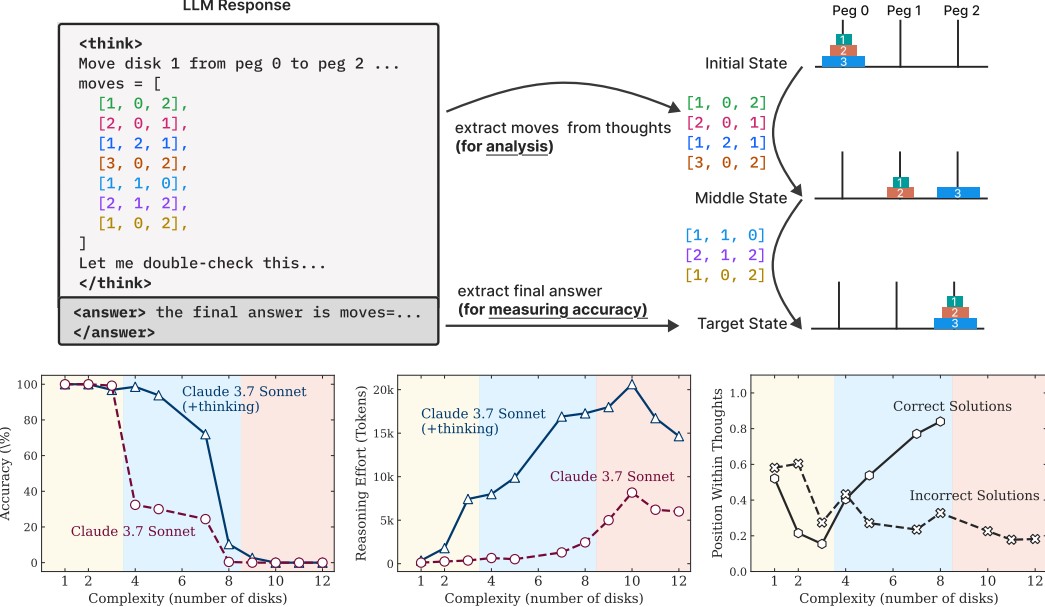

Figure 1: **Top**: Our setup enables verification of both final answers and intermediate reasoning traces, allowing detailed analysis of model thinking mechanisms. **Bottom left & middle**: At low complexity, non-thinking models are more accurate and token-efficient. As complexity increases, reasoning models outperform but require more tokens—until both collapse beyond a critical threshold, with shorter traces. **Bottom right**: For correctly solved cases, Claude 3.7 Sonnet (Thinking) tends to find answers early at low complexity and later at higher complexity. In failed cases, it often fixates on an early wrong answers, wasting the remaining token budget. Both cases reveal inefficiencies in the reasoning process.

self-reflection, and have demonstrated promising results across various reasoning benchmarks. Their emergence suggests a potential paradigm shift in how LLM systems approach complex reasoning and problem-solving tasks, with some researchers proposing them as significant steps toward more general artificial intelligence capabilities.

Despite these claims and performance advancements, the fundamental benefits and limitations of LRMs remain insufficiently understood. Critical questions still persist: Are these models capable of generalizable reasoning, or are they leveraging different forms of pattern matching [6]? How does their performance scale with increasing problem complexity? How do they compare to their standard LLM (non-reasoning) counterparts when provided with the same inference token compute? Most importantly, what are the inherent limitations of current reasoning approaches, and what improvements might be necessary to advance toward more robust reasoning capabilities?

We believe the lack of systematic analyses investigating these questions is due to limitations in current evaluation paradigms. Existing evaluations predominantly focus on established mathematical and coding benchmarks, which, while valuable, often suffer from data contamination issues and do not allow for controlled experimental conditions across different settings and complexities. Moreover, these evaluations do not provide insights into the structure and quality of intermediate reasoning traces. To understand the reasoning behavior of these models more systematically, we need environments that enable controlled experimentation.

In this study, we probe the reasoning mechanisms of frontier LRMs through the lens of problem complexity. Instead of the standard and common math benchmarks, we adopt controllable puzzle environments that let us vary complexity systematically—by adjusting puzzle elements while preserving the core logic—and inspect both solutions and internal reasoning (Fig. 1, top). These puzzles: (1) offer fine-grained control over complexity; (2) require only the explicitly provided rules, emphasizing algorithmic reasoning; (3) create new controlled setting that avoid contamination common in established benchmarks; and (4) support rigorous, simulator-based evaluation, enabling precise solution checks and detailed failure analyses.

Our empirical investigation reveals several key findings about current Language Reasoning Models (LRMs): First, despite their sophisticated self-reflection mechanisms learned through reinforcement learning, we observe that these models still fail to develop generalizable problem-solving capabilities for planning tasks, with performance collapsing to near-zero beyond a certain complexity threshold. Second, our comparison between LRMs and standard LLMs under equivalent inference token compute reveals three distinct reasoning regimes (Fig. 1, bottom). For simpler, low-compositional problems, standard LLMs demonstrate greater efficiency and accuracy. As problem complexity moderately increases, reasoning (thinking) models gain advantage. However, when problems reach high complexity with longer compositional depth, both model types experience complete performance collapse (Fig. 1, bottom left). Counterintuitively, near this collapse point, LRMs begin reducing their reasoning effort (measured by inference-time thinking tokens) as problem complexity increases, despite operating well below generation length limits (Fig. 1, bottom middle). This suggests a fundamental inference-time scaling limitation in LRMs' reasoning capabilities relative to problem complexity. Finally, our analysis of intermediate reasoning traces or thoughts reveals complexity-dependent patterns: In simpler problems, reasoning models often identify correct solutions early but inefficiently continue exploring incorrect alternatives—an "overthinking" phenomenon. At moderate complexity, correct solutions emerge only after extensive exploration of incorrect paths. Beyond a certain complexity threshold, models completely fail to find correct solutions and fixate on early incorrect attempts which leads to wasting the remaining inference token budget (Fig. 1, bottom right). This indicates LRMs possess limited self-correction capabilities that, while valuable, show clear inefficiencies and scaling limitations.

These findings highlight both the strengths and limitations of existing LRMs, raising questions about the nature of reasoning in these systems with important implications for their design and deployment. Our key contributions are as follows:

- We question the current evaluation paradigm of reasoning models on established math benchmarks and design a controlled experimental testbed by leveraging algorithmic puzzle environments that enable controllable experimentation with respect to problem complexity.

- We show that frontier LRMs (e.g., o3-mini, DeepSeek-R1, Claude-3.7-Sonnet-Thinking) still fail to develop generalizable problem-solving capabilities, with accuracy ultimately collapsing to near-zero beyond certain complexities across different environments.

- We find that there exists a scaling limit in the LRMs' reasoning effort with respect to problem complexity, evidenced by the counterintuitive decreasing trend in the thinking tokens after specific complexity points.

- We question the current evaluation paradigm based on final accuracy and extend our evaluation to intermediate reasoning traces using rigorous puzzle simulators. Our findings show that as complexity increases, LRMs locate correct solutions later than incorrect ones within their thought, but beyond a complexity point, they fixate on early errors and cannot recover. This provides quantitative insights into LRMs' self-correction mechanism and its scaling limits with complexity.

- We uncover surprising behaviors in LRMs' ability to perform exact computation, including their failure to benefit from explicit algorithms and their inconsistent reasoning across problems and scales.

## 2   Related Works

**Reasoning in Language Models.**   Large Language Models (LLMs) undergo multiple costly training phases using vast amounts of training data. While these LLMs demonstrate promising language understanding with strong compression capabilities, their intelligence and reasoning abilities remain a critical topic of scientific debate [7, 8]. Earlier iterations of LLMs [9–11] exhibited poor performance on reasoning benchmarks [12–14, 6]. To address these shortcomings, several approaches have been explored with the common theme among them being *"scaling"* both the training data and test-time computation. For instance, generating a Chain of Thought (CoT) [15–18] and incorporating self-verification [19–21] prior to the final answer have been shown to improve model performance. However, obtaining high-quality and scalable CoT data is quite expensive due to its scarcity. Another line of research focuses on compensating for the lack of supervised data by teaching models to think more effectively through supervised learning or reinforcement learning [22–27]. A notable open-source example of these improvements is Deepseek-R1 [3], which demonstrated that applying RL with verifiable rewards can significantly enhance model reasoning performance, matching that of closed models like OpenAI's o1 [2], leading to a new generation of language models referred to as Large Reasoning Models (LRMs) such as Gemini flash thinking [5], Claude Sonnet thinking [4], etc.

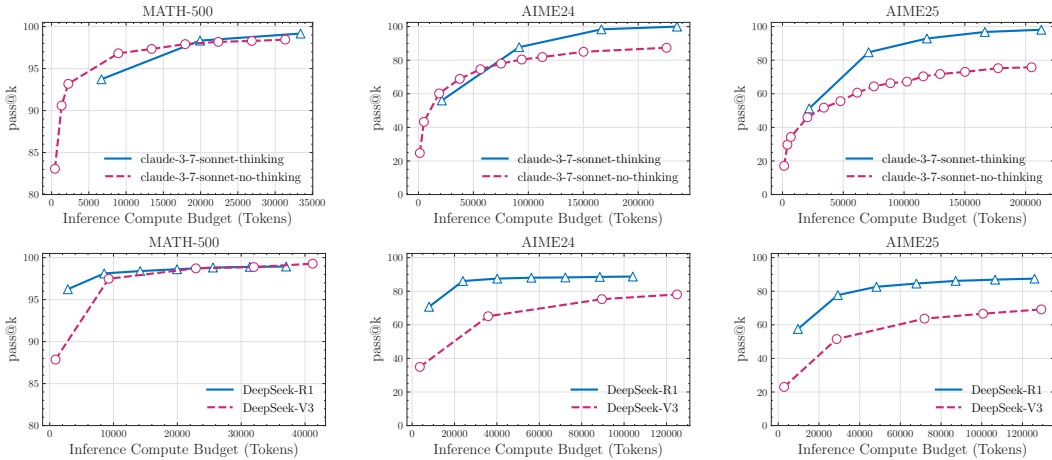

Figure 2: Comparative analysis of reasoning (thinking) versus non-reasoning (no-thinking) model variants across math benchmarks reveals inconsistent performance patterns.

**Understanding Large Reasoning Models.** Recent studies have explored various aspects of reasoning behavior: Large Reasoning Models have shown emergent behaviors such as discrepancy between thought traces and final answers [28, 29] as well as efficiency concerns through what researchers term the *"overthinking phenomenon"* [30–33], where models produce verbose, redundant outputs, even after finding the solution, creating significant inference computational overhead. In this work, we systematically analyze how much the model thinks with respect to task complexity. Recent works [34, 33, 32] have demonstrated that in newer LRMs accuracy generally declines when thinking increases in math problems, in contrast we observe that the opposite correlation of thinking and accuracy actually depends on the problem complexity and it only happens up to some complexity threshold in controlled puzzle environments. Yue et al. [35] have questioned whether reinforcement learning elicits novel reasoning patterns, observing that the pass@k performance of reasoning and non-reasoning models ultimately converge with larger samples. We also observe that in math datasets like MATH-500, pass@k is close for reasoning versus non-reasoning models but we observed different patterns under medium and high complexity of puzzles, which is not easily observable on established math benchmarks used in common evaluations.

**Controllable Evaluation Environments.** Unlike earlier studies that are mostly focused on mathematical benchmarks to evaluate the reasoning capabilities of language models, this work introduces controllable puzzle environments. These environments allow for precise manipulation of problem complexity while maintaining consistent logical processes, enabling a more rigorous analysis of reasoning patterns and limitations. Controllable environments are not uncommon in the relevant theoretical [36] and empirical studies in recent literature [12, 37–39]. However, our primary aim is not to propose a new benchmark; instead, we use similar benchmarks as tools for designing experiments to better understand the reasoning behavior of language models. Two closely related studies by Valmeekam et al. [40] and Ruoss et al. [39] demonstrated that reasoning models such as o1 show significant performance improvements compared to previous models on puzzles and decision-making tasks. Our work offers additional insights, such as examining pairs of reasoning/non-reasoning models (e.g., DeepSeek-R1/V3, Claude 3.7 Sonnet thinking/non-thinking). We also study the reasoning traces of the LRMs in more depth, revealing different complexity-dependent behaviors. Overall, the promising results from recent LRMs raise a critical question: how much have the previously reported limitations of LLMs been improved? In this work, we move beyond only measuring the final accuracy and analyze how well these LRMs tackle problems of varying complexities with controllable experiments, examining the properties of their reasoning processes.

## 3 Math and Puzzle Environments

Currently, it is not clear whether the performance enhancements observed in recent RL-based reasoning (thinking) models are attributable to increased exposure to established mathematical reasoning data, to the significantly greater inference compute allocated to longer thinking tokens, or to reasoning capabilities developed by RL-based training? Recent studies [35, 41] have explored this question with established math benchmarks by comparing the upper-bound capabilities (pass@k)

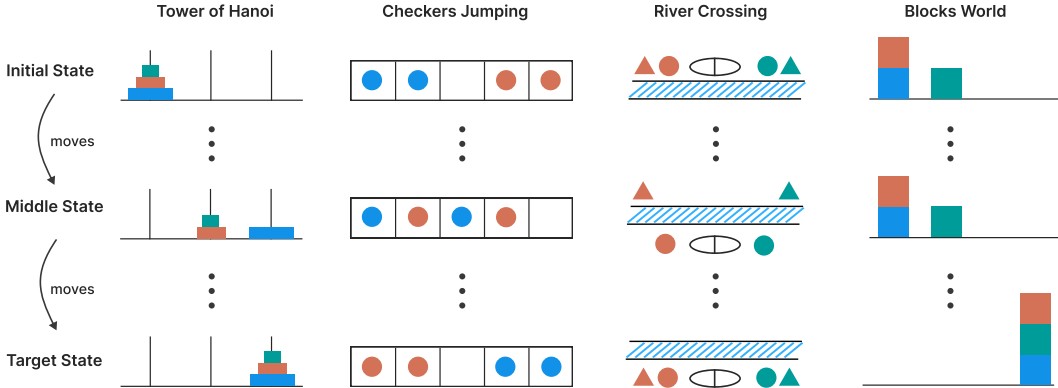

Figure 3: Illustration of the four puzzle environments. Rows show the progression from **initial state (top)** through **intermediate state (middle)** to **target state (bottom)** for puzzles: Tower of Hanoi (disk transfer across pegs), Checkers Jumping (position swapping of colored tokens), River Crossing (transporting entities across a river), and Blocks World (stack reconfiguration).

of RL-based reasoning (thinking) models with their non-reasoning (non-thinking) standard LLM counterparts. They have shown that under equivalent inference token budgets, non-thinking LLMs can eventually reach performance comparable to thinking models on benchmarks like MATH500 [42] and AIME24 [43]. We also conducted our comparative analysis of frontier LRMs like *Claude-3.7-Sonnet (with vs. without thinking)* and *DeepSeek (R1 vs. V3)*. Our results (shown in Fig. 2) confirm that, on the MATH500 dataset, the pass@k performance of thinking models is comparable to their non-thinking counterparts when provided with the same inference token budget. However, we observed that this performance gap slightly widens on the AIME24 benchmark and widens further on AIME25. This widening gap presents an interpretive challenge. It could be attributed to either: (1) increasing complexity requiring more sophisticated reasoning processes, thus revealing genuine advantages of the thinking models for more complex problems, or (2) reduced data contamination in newer benchmarks (particularly AIME25). Interestingly, human performance on AIME25 was actually higher than on AIME24 [44, 45], suggesting that AIME25 might be less complex. Yet models perform worse on AIME25 than AIME24—suggesting potential for some degree of data contamination in the training of frontier LRMs. Given these non-justified observations and the fact that mathematical benchmarks do not allow for controlled experimentation and manipulation of complexity, we turned to puzzle environments that enable more precise and systematic experimentation.

### 3.1 Puzzle Environments

We evaluate LRM reasoning on four controllable puzzles spanning compositional depth, planning complexity, and distributional settings. The puzzles are defined below and a schematic illustration is provided in Fig. 3.

**Tower of Hanoi** is a puzzle featuring three pegs and $n$ disks of different sizes stacked on the first peg in size order (largest at bottom). The goal is to transfer all disks from the first peg to the third peg. Valid moves include moving only one disk at a time, taking only the top disk from a peg, and never placing a larger disk on top of a smaller one. The difficulty in this task can be controlled by the number of initial disks as the minimum number of required moves with $n$ initial disks will be $2^n - 1$. However, in this work we do not grade for optimality of final solution and only measuring the correctness of each move and reaching the target state.

**Checker Jumping** is a one-dimensional puzzle arranging red checkers, blue checkers, and a single empty space in a line. The objective is to swap the positions of all red and blue checkers, effectively mirroring the initial configuration. Valid moves include sliding a checker into an adjacent empty space or jumping over exactly one checker of the opposite color to land in an empty space. No checker can move backward in the puzzle process. The complexity of this task can be controlled by the number of checkers: with $2n$ checkers, the minimum number of moves required will be $(n + 1)^2 - 1$.

**River Crossing** is a constraint satisfaction planning puzzle involving $n$ actors and their corresponding $n$ agents who must cross a river using a boat. The goal is to transport all $2n$ individuals from the left bank to the right bank. The boat can carry at most $k$ individuals and cannot travel empty. Invalid situations arise when an actor is in the presence of another agent without their own agent present, as

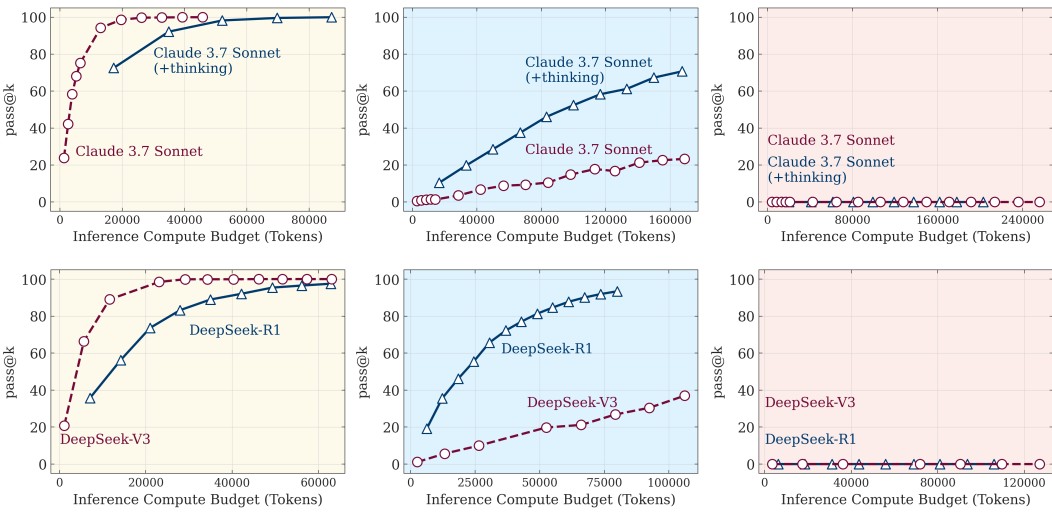

Figure 4: Pass@k performance of thinking models (Claude 3.7 Sonnet with extended thinking, DeepSeek-R1) versus their non-thinking counterparts (Claude 3.7 Sonnet, DeepSeek-V3) across equivalent inference compute budgets in puzzle environments of low , medium , and high complexity. Non-thinking models outperform in simple problems, thinking models show advantages at medium complexity, while both approaches fail at high complexity regardless of compute allocation.

each agent must protect their client from competing agents. The complexity of this task can also be controlled by the number of actor/agent pairs present. For $n = 2, n = 3$ pairs, we use boat capacity of $k = 2$ and for larger number of pairs we use $k = 3$.

**Blocks World** is a block-stacking puzzle requiring rearrangement of blocks from an initial configuration into a specified goal configuration. The objective is to find the minimum number of moves needed for this transformation. Valid moves are restricted to the topmost block of any stack, which can be placed either on an empty stack or on top of another block. The complexity in this task can also be controlled by the number of blocks present.

## 4 Experiments & Results

### 4.1 Experimental Setup

Most of our experiments are conducted on reasoning models and their non-reasoning counterparts, such as Claude 3.7 Sonnet (thinking/non-thinking) and DeepSeek-R1/V3. We chose these models because they allow access to the thinking traces, unlike models such as OpenAI's o-series. For experiments focused solely on final accuracy, we also report results on the o3-mini model. For Claude 3.7 Sonnet models, we allow the maximum token budget ($64$k). Similarly, for DeepSeek-R1/V3 models on local servers, we allow the maximum length to be up to $64$k tokens. For each puzzle instance and complexity level, we analyze 25 samples per model. We apply a filtering process to ensure the analyzed samples follow the requested response format (including sequence of moves in the specified format, reasoning steps, etc.). Comprehensive details of our experimental setup and results are provided in the Appendix.

### 4.2 How Does Complexity Affect Reasoning?

#### 4.2.1 Three Regimes of Complexity

Motivated by the observations in Fig. 2, to systematically investigate the impact of problem complexity on reasoning behavior, we conducted experiments comparing **thinking** (reasoning model with long CoT enabled by RL) and **non-thinking** (standard) model pairs across our controlled puzzle environments. Our analysis focus on matched pairs of LLMs with same model backbones, specifically *Claude-3.7-Sonnet (w. vs. w/o thinking)* and *DeepSeek (R1 vs. V3)*. In each puzzle, we vary the complexity by manipulating problem size $N$ (representing disk count, checker count, block count, or crossing elements).

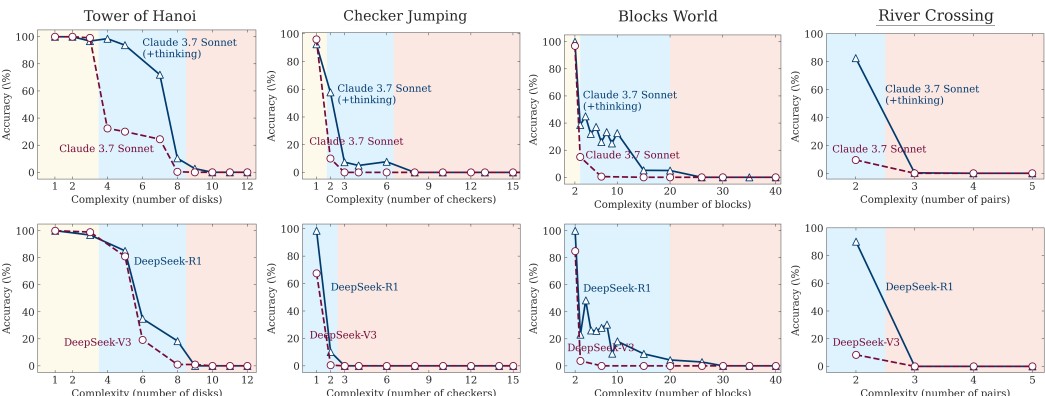

Figure 5: Accuracy of thinking models (Claude 3.7 Sonnet with extended thinking, DeepSeek-R1) versus their non-thinking counterparts (Claude 3.7 Sonnet, DeepSeek-V3) across all puzzle environments and varying levels of problem complexity.

Fig. 4 shows the upper bound performance capabilities (pass@k) of these model pairs under equivalent inference token compute (averaged across all puzzles), extending earlier analyses from mathematical benchmarks (Fig. 2) to the controlled puzzle environments. Complementing this, Fig. 5 presents the accuracy of both model types as a function of problem complexity across each puzzle environment. Results from both these figures demonstrate that, unlike observations from math, there exists *three regimes* in the behavior of these models with respect to complexity. In the first regime where problem complexity is low, we observe that non-thinking models are capable of obtaining performance comparable to, or even better than thinking models with more token-efficient inference. In the second regime with medium complexity, the advantage of reasoning models capable of generating long chain-of-thought begin to manifest, and the performance gap between model pairs increases. The most interesting regime is the third regime where problem complexity is higher and the performance of both models have collapsed to zero. Results show that while thinking models delay this collapse, they also ultimately encounter the same fundamental limitations as their non-thinking counterparts.

Additional results comparing the *QwQ-32B* and *Qwen2.5-32B* model pairs under the same thinking vs. non-thinking setup are provided in Appendix A.6.

### 4.2.2 Collapse of Reasoning Models

We next examine how different specialized reasoning models equipped with thinking tokens respond to increasing problem complexity. Our experiments evaluate five thinking models: *o3-mini* (medium and high configurations), *DeepSeek-R1*, *DeepSeek-R1-Distill-Qwen-32B*, and *Claude-3.7-Sonnet (thinking)*. Fig. 6 demonstrates these models' performance in terms of accuracy (top) and thinking token usage (bottom) across varying complexity levels. Results show that all reasoning models exhibit a similar pattern with respect to complexity: accuracy progressively declines as problem complexity increases until reaching collapse (near-zero accuracy) beyond a model-specific complexity threshold. Analysis of inference thinking token compute also reveals an intriguing pattern in thinking token allocation learned by these models. We observe that reasoning models initially increase their thinking tokens proportionally with problem complexity. However, upon approaching a critical threshold—which closely corresponds to their accuracy collapse point—models counterintuitively begin to reduce their reasoning effort despite increasing problem complexity. This phenomenon is most pronounced in o3-mini variants and less severe in the Claude-3.7-Sonnet (thinking) model. Notably, despite mostly operating well below their generation length limits with ample inference budget available these models fail to take advantage of additional inference compute during the thinking phase as problems become more complex. This behavior suggests a fundamental scaling limitation in the thinking capabilities of current reasoning models relative to problem complexity.

### 4.3 What Happens Inside the Thoughts of Reasoning Models?

To gain deeper insights into the thinking processes of reasoning models, we conducted a fine-grained analysis of their reasoning traces. As shown in Fig. 1, our setup with puzzle environments allows us to look beyond final answer and obtain more detailed insight into the reasoning traces ("thoughts")

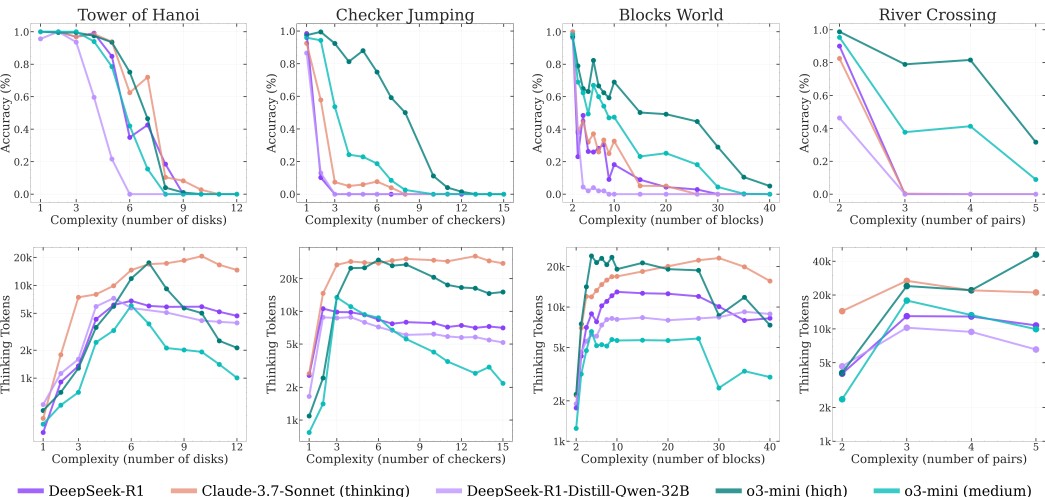

Figure 6: Accuracy and thinking tokens versus problem complexity for reasoning models across puzzle environments. As complexity increases, reasoning models initially spend more tokens while accuracy declines gradually, until a critical point where reasoning collapses—performance drops sharply and reasoning effort counterintuitively decreases.

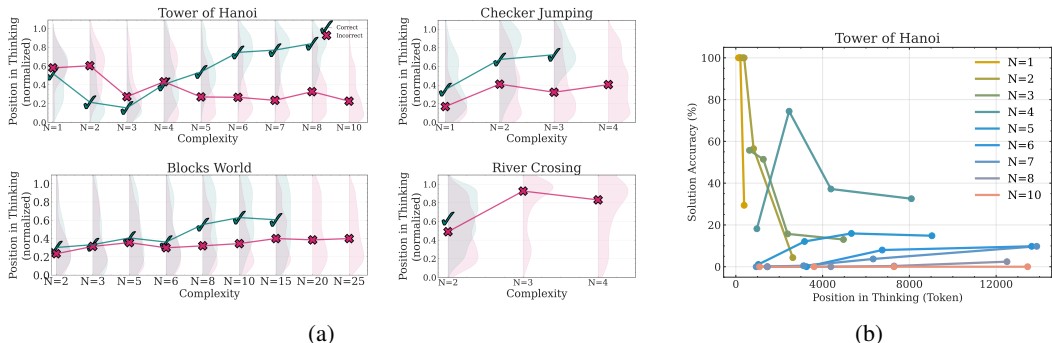

(a)                                                                                  (b)

Figure 7: **(a)** Position and correctness of intermediate solutions within reasoning traces across four puzzles at varying complexity levels. ✓ indicates correct solutions, ✗ indicates incorrect solutions, with distribution density shown by shading; **(b)** Solution accuracy versus position in thinking for Tower of Hanoi at different complexity levels. Simple problems (N=1-4) show early accuracy declining over time (overthinking), moderate problems (N=5-7) show slight improvement in accuracy with continued reasoning, and complex problems (N≥8) exhibit consistently near-zero accuracy, indicating complete reasoning failure.

produced by these models. We extract and analyze the intermediate solutions explored *within the thoughts* of a model with the help of puzzle simulators. Our investigation examines the patterns and characteristics of these intermediate solutions, their correctness relative to their sequential position in the reasoning process, and how these patterns evolve with increasing problem complexity. For this analysis, we focus on the reasoning traces generated by *Claude-3.7-Sonnet-Thinking* across our puzzle suite. For each unique intermediate solution identified within the traces, we recorded: (1) its relative position within the reasoning trace (normalized by total thought length), (2) its correctness and specific failure move as validated by our puzzle simulators, and (3) the complexity of the corresponding problem. This allows to characterize the progression and accuracy of solution development throughout the reasoning process.

Fig. 7a demonstrates the relation between the position of intermediate solutions within thoughts, their correctness, and problem complexity across all puzzle environments. Our analysis from intermediate reasoning traces also further validates three regimes of complexity discussed above. For simpler problems, reasoning models often find the correct solution early in their thinking but then continue exploring incorrect solutions. Note the distribution of incorrect solutions (red) is comparable or shifted more upward towards end of thinking compared to correct solutions (green). This phenomenon,

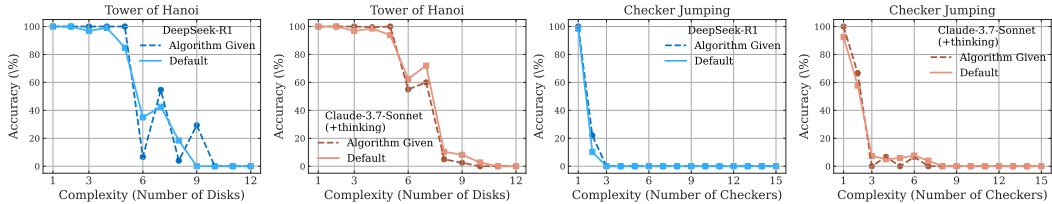

Figure 8: Performance comparison between default problem-solving and algorithm-guided execution across Tower of Hanoi and Checker Jumping puzzles. Even when given the solution algorithm and only needing to execute prescribed steps, failures occur at similar points, highlighting reasoning models' limitations in following logical procedures.

referred to as "overthinking" in the literature, leads to the waste of compute. As problems become moderately more complex, this trend reverses: models first explore incorrect solutions and mostly later in thought arrive at the correct ones. This time the distribution of incorrect solutions (red) is shifted more downward compared to correct ones (green). Finally, for the problems with higher complexity, collapse emerges, meaning that the model fails to recover any correct solutions within the thought and it often fixates on an early incorrect solutions, wasting the remaining thought token budget.

Fig. 7b presents a complementary analysis of solution accuracy within sequential segments (bins) of the thoughts in the Tower of Hanoi environment. It can be observed that for simpler problems (smaller N), solution accuracy tends to decrease or oscillate as thinking progresses, providing further evidence of the overthinking phenomenon. However, this trend changes for more complex problems, where solution accuracy increases with thinking progression—up to a certain threshold. Beyond this complexity threshold, in the "collapse mode", accuracy is zero and model fail to recover any incorrect solution.

## 4.4 Open Questions: Puzzling Behavior of Reasoning Models

In this section, we present surprising results concerning the limitations of reasoning models in executing exact problem-solving steps, as well as demonstrating different behaviors of the models based on the failure moves. As shown in Figure 8, in the Tower of Hanoi and Checker Jumping puzzle environments, even when we provide the algorithm in the prompt—so that the model only needs to execute the prescribed steps—performance does not improve, and the observed collapse still occurs at roughly the same point on both puzzle environments. This is noteworthy because finding and devising a solution should require substantially more computation (e.g., for search and verification) than merely executing a given algorithm. This further highlights the limitations of reasoning models in verification and in following logical steps to solve a problem, suggesting that further research might be needed to better understand the symbolic manipulation capabilities of such models [46, 6].

Moreover, by looking deeper into failure cases, shown in Figure 9 for Claude-3.7-Sonnet thinking as well as Figures 14 and 15 in Appendix for other models, we observe inconsistencies in how models apply learned solution strategies across different problems and scales. Sometimes, models exhibit a non-monotonic failure behavior with respect to problem complexity—instances where models fail earlier in the solution sequence for higher $N$ values despite requiring longer overall solutions. For example, Figure 9 shows that in Tower of Hanoi, failure happens at below 50 moves for $N = 12$ but the model succeeds through more than 100 moves for $N = 10$, contradicting the expectation that effective algorithmic planning and execution for the same puzzle should maintain consistent failure patterns. Also, we observe longer error-free sequences in some puzzle environments compared to others. For example, in the Tower of Hanoi environment, the model's first error in the

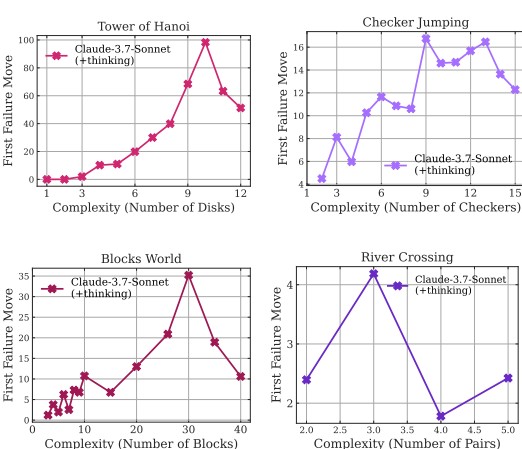

Figure 9: First failure move versus problem complexity for Claude 3.7 Sonnet (thinking) across four puzzle environments.

proposed solution often occurs much later, e.g., around move 100 for (N=10), compared to the River Crossing environment, where the model can only produce a valid solution until move 4. Note that this model also achieves near-perfect accuracy when solving the Tower of Hanoi with (N=5), which requires 31 moves, while it fails to solve the River Crossing puzzle when (N=3), which has a solution of 11 moves. Although the branching factor for solution exploration in River Crossing is larger than in Tower of Hanoi, this analysis on the comparison of computational complexity between puzzles is asymptotic and doesn't hold for the small values of N in our experiments where collapse happens. The search space for a valid 11-move (N=3) River Crossing solution is vastly smaller than the search space for a 255-move (N=8) Tower of Hanoi solution where models begin to fail. This likely suggests that examples of River Crossing puzzle with larger N are less familiar for the model, meaning LRMs may not have frequently encountered such instances during training. At the end, LLMs (or LRMs) are complex artifacts and we cannot easily say which problem is easier or more complex for them only based on computational complexity and without knowing about their training data distribution. How they approach complexity does not necessarily corresponds to the actual computational complexity of the problem and more to the learned solution distributions. That's why our focus on complexity is mostly to track model behavior within each puzzle setting rather than between the puzzles.

## 5    Conclusion

In this paper, we systematically examine frontier Large Reasoning Models (LRMs) through the lens of problem complexity using controllable puzzle environments. Our findings reveal several strengths and limitations in current models: despite sophisticated self-reflection mechanisms learned by reinforcement learning, these models still fail to develop generalizable reasoning capabilities beyond certain complexity thresholds. We identified three distinct reasoning regimes: standard LLMs outperform LRMs at low complexity, LRMs show advantage at moderate complexity, and both collapse at high complexity. We observe the counterintuitive reduction in reasoning effort as problems approach critical complexity, suggesting an inherent compute scaling limit in LRMs. Our detailed analysis of reasoning traces further exposed complexity-dependent reasoning patterns, from inefficient "overthinking" on simpler problems to complete failure on complex ones. These insights help us to better understand the nature reasoning in LRMs and challenge some of the prevailing assumptions about LRM's capabilities. Finally, we presented some of our surprising observations on LRMs that lead to several open questions for future research. Most notably, we observed their limitations in performing exact computation; for example, when we provided the solution algorithm of puzzle to the models, their performance on this puzzle did not improve. Investigating the first failure move of the models also revealed some surprising and inconsistent behaviors. Our results show that models sometimes fail earlier on harder instances despite requiring longer overall solutions. They also show very different error-free sequence lengths across puzzles, e.g., performing up to 100 correct moves in Tower of Hanoi but only 4 in River Crossing. This disparity likely reflects differences in the models' familiarity with each setting based on the distribution of training data rather than intrinsic differences in problem's computational complexity. We hope that these findings can pave the way for future investigations into better understanding the reasoning process of these systems.

**Limitations**

We acknowledge that our work has limitations. While our puzzle environments enable controlled experimentation with fine-grained control over problem complexity, they represent a narrow slice of reasoning tasks and may not capture the diversity of real-world or knowledge-intensive reasoning problems. It is notable that most of our experiments rely on black-box API access to the closed frontier LRMs, limiting our ability to mechanistically analyze internal states or architectural components. Furthermore, the use of deterministic puzzle simulators assumes that reasoning can be perfectly validated step by step. However, in less structured domains, such precise validation may not be feasible, limiting the transferability of this analysis to other more generalizable and open-ended reasoning.

**Acknowledgments**

The authors would like to thank Scott Hoang, Yichen Jiang, Minsik Cho, Mohammad Sekhavat, David Harrison, Mohammadreza Armandpour and Devi Krishna for the valuable feedback and support.

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

# A Appendix

In this appendix, we provide details supplementing the main text, including our response to alternative critics, experimental setup specifications, additional results, and extended analysis.

## A.1 Response to Main Criticisms

This section addresses critiques raised regarding our paper and provides our responses to these concerns. We appreciate the engagement from the research community and have incorporated several modifications to address valid points while clarifying misunderstandings where appropriate.

**Question: Are failures on Tower of Hanoi due to context limit issues rather than reasoning limitations?**

**Response:** Our empirical evidence demonstrates that model failures observed in experiments occur within the context limits in all puzzles. For the Tower of Hanoi, with its exponential growth raising questions of context-limit failures, reasoning models begin to collapse at N=7 and 8, corresponding to ~100-200 moves (as shown in Figures 5, 6 and 11) which is well within the context limits. More importantly, if we look deeper into the failure cases (like in Figures 9, 14, and 15), we see that the first failure move actually happens much sooner than the final move. For example, for Tower of Hanoi with N≈10 (requiring $\sim 10^3$ moves), failure typically occurs within the first ~100 moves (10% of solution length); for N=8, which requires 255 moves, failure occurs around ~40 moves (15% of solution length). This indicates that model failures happen much earlier and are not due to context limits. To account for context limit concerns for large values of N in Tower of Hanoi, we have removed N>12 from the experiments on this puzzle.

**Question: Do models collapse primarily due to the sampling of large number of moves (particularly on Tower of Hanoi)?**

**Response:** Some critics argue that sampling is the primary cause of the observed collapse behavior (particularly on Tower of Hanoi). While we acknowledge that longer move sequences may reduce the chance of correct reasoning chains due to sampling, our experiments demonstrate that sampling is not the primary factor underlying these failures. To investigate this, we conducted additional ablation experiments using temperature zero to remove sampling effects on solution length. These experiments were performed on the DeepSeek-R1 reasoning model to ensure precise temperature control on local servers. Results show that eliminating sampling does not change the collapse across any of the puzzles, including Tower of Hanoi, indicating that sampling of long sequences is not causing the observed collapse behavior. In fact, in Tower of Hanoi and Blocks World, sampling actually helps delay collapse. For example, in Blocks World, R1 with temperature zero collapses at N=4, but sampling delays the collapse until N=30 (details in Figure 13 of Appendix A.6). Similar collapse behavior in much shorter sequences where sampling has less impact like River Crossing (11 moves, N=3) and Checker Jumping (24 moves, N=4) also further supports this conclusion.

**Question: Does River Crossing with N≥6 invalidate our core findings on this puzzle?**

**Response:** Upon further experiments, we found that the puzzle dynamics shift considerably for N≥6, where the optimal boat capacity ($k = 4$) fundamentally changes the problem structure, making it less suitable for evaluating planning capabilities. This insight led us to refine our analysis by focusing on the cases where N<6. However, this modification does not invalidate our core findings, as performance of models mostly collapse earlier from N=3, which requires only an 11-move solution for basic constraint satisfaction. This early performance collapse is particularly noteworthy and suggests that current frontier reasoning models may have limited capability for constraint verification and satisfaction while planning.

**Question: Is the finding that providing algorithms doesn't improve performance specific to Tower of Hanoi due to its well-known recursive algorithm or reflecting a general limitation of reasoning models?**

**Response:** The critic suggests that the findings of algorithm execution limitation in LRMs (Section 4.4) may be only an artifact of the Tower of Hanoi's well-known recursive algorithm being already memorized by models, rather than indicating a general limitation in algorithm execution. To investigate this, we conducted additional experiments on Checker Jumping puzzle, with some different characteristics: it requires fewer moves than Tower of Hanoi, exhibits earlier collapse behavior, and

appears to have less algorithmic familiarity. If the critic's hypothesis were correct, we should observe clear benefits from algorithm provision in this problem. However, our results in Figure 8 show very similar patterns for both puzzles. Even when given explicit algorithm steps—requiring only execution rather than problem-solving and solution discovery—models show no meaningful improvement on either puzzle, with collapse happening at similar points. These results further validate our findings regarding limitations in the execution of logical steps and suggest a more fundamental failure mode in reasoning models, rather than one specific to the Tower of Hanoi.

**Question: Why not use tools for solving these puzzles?**

**Response:** We would like to highlight that our objective is to assess models' reasoning processes—their ability to understand problems, explore solution spaces, and execute logical steps—rather than merely achieving correct final answers. For the puzzles in our study, algorithmic solutions are very well-established and likely exist in models' training data as standard implementations. Allowing tool use would tell us nothing about their capacity for problem understanding, constraint reasoning, or multi-step logical execution. Tool usage becomes valuable when algorithmic codes for solving the problem are unknown and require compositional reasoning to be developed.

We believe it's crucial to separate tool use from understanding when evaluating reasoning capabilities. As the Chinese room argument illustrates, it's entirely possible to use tools effectively without any real comprehension of the underlying problem. Consider a student who complains about a math exam requiring integration by hand, arguing that software can produce correct answers instantly. The teacher's goal isn't to find the answer—they already know it—but to assess the student's conceptual understanding. The same principle applies to LLM evaluation. Furthermore, if an LLM cannot reliably perform sequential reasoning steps on its own, how can we expect it to write complex code or coordinate multiple tools correctly for even more challenging problems? Reliable tool use requires the same systematic thinking and logical execution that these simple puzzles assess.

## A.2 Details on Puzzle Environment Specifications and Design

### A.2.1 Tower of Hanoi

**Problem Description.** The Tower of Hanoi is a classic recursive puzzle that serves as a great problem for evaluating sequential reasoning and planning capabilities in reasoning models. The puzzle consists of three pegs (labeled 0, 1, and 2 from left to right) and $N$ disks of varying sizes, where each disk is uniquely numbered from 1 (smallest) to $N$ (largest). In the initial configuration, all $N$ disks are stacked on the leftmost peg (peg 0) in descending order of size, with the largest disk at the bottom and the smallest at the top. The remaining two pegs (1 and 2) are initially empty. The goal is to transfer all disks from peg 0 to peg 2, maintaining the same size ordering (largest at bottom, smallest at top). This puzzle is governed by three fundamental constraints: (1) *Single Disk Movement:* Only one disk may be moved at a time; (2) *Top Disk Access:* Only the topmost disk from any peg can be selected for movement; and (3) *Size Ordering Constraint:* A larger disk may never be placed on top of a smaller disk. This puzzle is a good evaluation testbed for reasoning and planning capabilities of models as it requires models to demonstrate key cognitive demands such as breaking down the problem into subproblems (recursive thinking), tracking multiple states and disk positions simultaneously (working memory management), adhering to movement rules and constraints while planning ahead (constraint satisfaction), and determining the correct order of operations to achieve the final goal (sequential planning).

The minimum number of moves required to solve the Tower of Hanoi recursive puzzle with $N$ disks is $2^N - 1$, making it an exponentially scaling problem. This property allows for fine-grained difficulty control by adjusting the problem size with number of initial disks. However, in our evaluation framework, we focus on solution correctness rather than optimality, assessing each of the move's validity and the model's ability to reach the target state as the success criteria.

**Prompt Design.** The system prompt begins with a clear problem statement describing the puzzle setup. It explicitly states the movement rules and the objective of transferring all disks to the third peg. To facilitate understanding, the prompt includes example demonstrations as well as the critical formatting and reasoning expectations.

1. Only one disk can be moved at a time.

2. Each move consists of taking the upper disk from one stack and placing it on top of another stack.

3. A larger disk may not be placed on top of a smaller disk.

This means: Move disk 1 from peg 0 to peg 2, then move disk 2 from peg 0 to peg 1, and so on.

The user prompt after the system prompt presents the specific puzzle instance with current configuration showing the distribution of disks across pegs and the goal configuration specifying the target state.

**Simulator.** Our evaluation framework employs separate puzzle simulators for each puzzle to ensure consistent assessment and rigorous failure analysis of solutions obtained from LRMs. The Tower of Hanoi simulator is designed as a stateful environment that tracks disk configurations across three pegs and validates each proposed move against the puzzle's fundamental constraints. The simulator architecture follows a modular design pattern with clear separation between state management, move validation, and solution verification. In this simulator, we have a puzzle class which tracks the current

disk configuration and enforces the puzzle's fundamental constraints. We also have a method to execute each move in the puzzle setup and perform four-layer validation: checking peg boundary conditions (0-2), verifying source pegs contain disks, confirming the specified disk is topmost, and enforcing the size ordering constraint that prevents larger disks from being placed on smaller ones. Upon successful validation, the method executes the disk transfer and updates the puzzle state. Finally, the complete solution validation is processed by sequentially processing move lists, and verifying goal state achievement.

### A.2.2 Checker Jumping

**Problem Description.** Checker Jumping is a one-dimensional constraint-satisfaction puzzle designed to test sequential reasoning, planning, and rule understanding capabilities. The puzzle consists of a linear arrangement of red checkers ('R'), blue checkers ('B'), and a single empty space ('_'). In the standard configuration, $N$ red checkers are positioned on the left side, followed by an empty space in the middle, and $N$ blue checkers on the right side, forming a linear board of length $2N + 1$. The objective is to swap the positions of all red and blue checkers, effectively mirroring the initial configuration, where red checkers end up on the right and blue checkers on the left. Movement in this puzzle is governed by two fundamental rules: (1) *Slide Movement:* A checker can slide forward into an adjacent empty space; and (2) *Jump Movement:* A checker can jump forward over exactly one checker of the opposite color to land in an empty space. Therefore, checkers cannot move backward toward their starting side—red checkers can only move rightward, and blue checkers can only move leftward from the initial configuration. This puzzle presents cognitive challenges that make it a good testbed for reasoning models. For example, models must demonstrate some aspect of spatial reasoning (tracking checker positions and possible moves), constraint satisfaction (adhering to movement rules during puzzle), lookahead planning (anticipating how current moves affect future possibilities towards goal), and state-space exploration (searching through possible move sequences to find a valid solution path).

The difficulty of the Checker Jumping puzzle scales with the number of checkers: with $N$ checkers of each color, the minimum solution requires $(N + 1)^2 - 1$ moves, creating a quadratic relationship between problem size and solution complexity. In our evaluation framework, we mainly focus on solution correctness rather than optimality, evaluating each move against the puzzle constraints and confirming that the final state matches the goal configuration. This approach allows us to precisely identify reasoning failures and constraint violations that might occur during the solution process.

**Prompt Design.** The system prompt begins with a clear problem statement describing the puzzle setup and movement rules. It explicitly states the objective and provides a concrete example with a small board configuration to illustrate how moves should be represented.

---

**System Prompt - Checker Jumping**

You are a helpful assistant. Solve this puzzle for me.
On a one-dimensional board, there are red checkers ('R'), blue checkers ('B'), and one empty space ('_'). A checker can move by either:

1. Sliding forward into an adjacent empty space, or

2. Jumping over exactly one checker of the opposite color to land in an empty space.

The goal is to swap the positions of all red and blue checkers, effectively mirroring the initial state.
**Example:** If the initial state is ['R', '_', 'B'], the goal is to reach ['B', '_', 'R']. Your solution should be a list of moves where each move is represented as [checker_color, position_from, position_to]. For example:

```
moves = [['R', 0, 1], ['B', 2, 0], ['R', 1, 2]]
```

This means: Move the red checker from position 0 to 1, then move the blue checker from position 2 to 0, and so on.
**Requirements:**

---

- When exploring potential solutions in your thinking process, always include the corresponding complete list of moves.
- The positions are 0-indexed (the leftmost position is 0).
- Ensure your final answer includes the complete list of moves for final solution in the format: `moves = [[checker_color, position_from, position_to], ...]`

The user prompt presents the specific puzzle instance with the initial board configuration, and the goal state.

---

**User Prompt Template for $N$ Checkers - Checker Jumping**

I have a puzzle with $2N+1$ positions, where $N$ red checkers ('R') on left, $N$ blue checkers ('B') on right, and one empty space ('_') in between are arranged in a line.
**Initial board:** R R ... R _ B B ... B
**Goal board:** B B ... B _ R R ... R
**Rules:**

- A checker can slide into an adjacent empty space.
- A checker can jump over exactly one checker of the opposite color to land in an empty space.
- Checkers cannot move backwards (towards their starting side).

Find the sequence of moves to transform the initial board into the goal board.

---

**Simulator.** Our evaluation framework employs a custom simulator for validating Checker Jumping puzzle solutions. The simulator implements a comprehensive validation system that enforces all puzzle constraints while tracking the state evolution throughout the solution path. The Checker Jumping simulator is designed as a stateful environment that tracks the position of all checkers and the empty space, validating each move of a given solution against the puzzle's movement rules. The simulator begins by validating that both the initial and goal states are generally well-formed, containing the same number of red and blue checkers and exactly one empty space. Then, each move is executed with a method that performs multi-layer validation: verifying position boundaries, confirming correct checker color at source, ensuring target positions are empty, and validating move types as either slides (distance=1) or jumps (distance=2). The simulator enforces directional constraints preventing backward movement (red checkers only move right, blue checkers only move left) and validates jump moves by confirming the presence of an opposite-colored checker in the middle position. Upon successful validation, the method executes the checker transfer by updating positions and clearing the source. Then, the complete move sequences are processed with final goal state verification.

### A.2.3 River Crossing

**Problem Description.** River Crossing is a constraint satisfaction planning puzzle that tests multi-agent coordination and constraint management. This puzzle is a generalization of classic problems such as the Missionaries and Cannibals problem and the Bridge and Torch problem, which have been widely studied in planning literature [47, 48]. The river crossing puzzle involves $N$ actors (denoted by $a_1, a_2, ..., a_N$) and their corresponding $N$ agents (denoted by $A_1, A_2, ..., A_N$) who must cross a river using a boat. In the initial state, all $2N$ individuals are on the left bank of the river. The goal is to transport everyone safely to the right bank. The puzzle operates under several key movement constraints: (1) *Boat Capacity Constraint:* The boat can carry at most $k$ individuals at a time, where $k$ is typically set to 2 for smaller puzzles ($N \leq 3$) and 3 for larger puzzles; (2) *Non-Empty Boat Constraint:* The boat cannot travel empty and must have at least one person aboard; (3) *Safety Constraint:* An actor cannot be in the presence of another agent unless their own agent is also present, as agents must protect their clients from competing agents. This safety constraint applies both on the banks and in the boat. This puzzle requires complex planning and state tracking as participants must carefully

coordinate their crossings while maintaining safety constraints at all times. The solver must reason through different combinations of individuals who can safely travel together, determine who should return with the boat after a crossing, and strategically plan a sequence that eventually brings everyone to the right bank without violating any constraints. The complexity of this task can be controlled by adjusting the number of actor-agent pairs, creating a scalable challenge for reasoning models.

**Prompt Design.**    The system prompt introduces the notation for representing actors and agents, establishes the solution format as a list of boat moves, and provides a simple example to demonstrate the format.

---

**System Prompt - River Crossing**

You are a helpful assistant. Solve this puzzle for me.
You can represent actors with a1, a2, ... and agents with A1, A2, ... . Your solution must be a list of boat moves where each move indicates the people on the boat. For example, if there were two actors and two agents, you should return:

```
moves=[["A2", "a2"], ["A2"], ["A1", "A2"], ["A1"], ["A1", "a1"]]
```

which indicates that in the first move, A2 and a2 row from left to right, and in the second move, A2 rows from right to left and so on.
**Requirements:**

- When exploring potential solutions in your thinking process, always include the corresponding complete list of boat moves.

- The list shouldn't have comments.

- Ensure your final answer also includes the complete list of moves for final solution.

---

The user prompt presents the specific puzzle instance with $N$ actor-agent pairs, and the boat capacity $k$, and the safety constraint that must be maintained throughout the solution.

---

**User Prompt Template for $N$ Pairs - River Crossing**

$N$ actors and their $N$ agents want to cross a river in a boat that is capable of holding only $k$ people at a time, **with the constraint that no actor can be in the presence of another agent, including while riding the boat, unless their own agent is also present**, because each agent is worried their rivals will poach their client. Initially, all actors and agents are on the left side of the river with the boat. How should they cross the river? (Note: the boat cannot travel empty)

---

**Simulator.**    Our evaluation framework employs a custom simulator for validating River Crossing puzzle extracted solutions. The simulator tracks the state of all individuals (actors and agents) and the boat position while enforcing all puzzle constraints. Each move is executed with multi-step validation: checking boat capacity limits, verifying all passengers are on the boat's current side, and enforcing the critical safety constraint that actors cannot be in the presence of other agents without their own agent present, both on the boat and on each bank after the move. The simulator manages dynamic boat positioning, automatically switching sides after each crossing, and validates the complete state after each move to ensure no safety violations occur on either bank. Then, the complete crossing sequences are verified that all $2N$ individuals successfully reach the right bank.

### A.2.4   Blocks World

**Problem Description.**    Blocks World is a classical planning puzzle that has been recently studied for analyzing the planning capabilities of LLMs [38, 40]. The puzzle involves multiple stacks of blocks (A, B, C, etc.) that must be rearranged from an initial configuration to a specified goal configuration. Each block is uniquely identified by its letter, and the objective is to find the sequence of valid moves

needed to transform the initial state exactly into the goal state. The puzzle operates only under two fundamental constraints: (1) *Top Block Movement:* Only the topmost block from any stack can be moved; and (2) *Valid Placement:* A block can only be placed either on an empty position or on top of another block. These constraints create planning problem where the order of operations becomes critical, as some configurations may require temporary placement of blocks to access those beneath them later. Blocks World serves as a good testbed for evaluating planning capabilities in reasoning models because it requires forward thinking, and state tracking. Recent studies have examined this puzzle in various configurations, including simplified settings with as few as 3 to 5 blocks, to evaluate LLM performance on sequential planning tasks [38, 40, 49]. Models must demonstrate the ability to decompose complex state transformations into valid sequential moves, reason about dependencies between blocks (e.g., unblocking lower blocks before accessing them), and efficiently plan paths to the goal state without illegal moves. Our design of this puzzle is motivated by recent works in literature [40, 49], but it's set to be more challenging for the recent reasoning models requiring more disassembly and reassembly of the stacks (eg. alternating between blocks from different stacks to reach target).

The difficulty of this puzzle can be scaled by adjusting several parameters: the number of blocks, the number of stacks, and the complexity of the initial and goal configurations. We primarily control complexity through the block count $N$, while following clear structural patterns in the initial and goal configurations. In our experimental design, the initial configuration consistently divides the $N$ blocks between two stacks in alphabetical order, with the third stack empty as workspace. The goal configuration consolidates all blocks onto the first stack in a systematic interleaved pattern that alternates between blocks from the two initial stacks, with specific positioning that requires complete disassembly and reassembly of the existing stacks. For example, for $N = 4$, the initial state has blocks divided between two stacks [["A", "B"], ["C", "D"], []] and the goal state [["D", "B", "C", "A"], [], []] requires interleaving blocks from both stacks; and for $N = 6$, the initial state [["A", "B", "C"], ["D", "E", "F"], []] must be transformed to [["F", "C", "E", "B", "D", "A"], [], []], forming a complex alternating pattern. As $N$ increases, the state space grows factorially, and the minimum solution length increases approximately linearly.

**Prompt Design.** The system prompt introduces the fundamental rules of the Blocks World puzzle, establishes the move representation format, and provides a simple example to demonstrate the solution structure.

---

**System Prompt - Blocks World**

You are a helpful assistant. Solve this puzzle for me.
In this puzzle, there are stacks of blocks, and the goal is to rearrange them into a target configuration using a sequence of moves where:

- Only the topmost block from any stack can be moved.
- A block can be placed either on an empty position or on top of another block.

**Example:** With initial state [["A", "B"], ["C"], []] and goal state [["A"], ["B"], ["C"]], a solution might be:

```
moves = [["C", 1, 2], ["B", 0, 1]]
```

This means: Move block C from stack 1 to stack 2, then move block B from stack 0 to stack 1.
**Requirements:**

- When exploring potential solutions in your thinking process, always include the corresponding complete list of moves.
- The positions are 0-indexed (the leftmost position is 0).
- Ensure your final answer also includes the complete list of moves for final solution in the format: `moves = [[block, from stack, to stack], ...]`

---

The user prompt presents the specific puzzle instance with the initial and goal configurations provided, and explicitly reminds the model about the movement constraint.

> **User Prompt Template for $N$ Blocks - Blocks World**
>
> I have a puzzle with $N$ blocks.
> **Initial state:**
>
> > Stack 0: $blocks\_0$ (top)
> >
> > Stack 1: $blocks\_1$ (top)
> >
> > ...
> >
> > Stack $m$: $blocks\_m$ (top)
>
> **Goal state:**
>
> > Stack 0: $goal\_blocks\_0$ (top)
> >
> > Stack 1: $goal\_blocks\_1$ (top)
> >
> > ...
> >
> > Stack $m$: $goal\_blocks\_m$ (top)
>
> Find the sequence of moves to transform the initial state into the goal state. Remember that only the topmost block of each stack can be moved.

**Simulator.** Our evaluation framework employs a custom simulator for validating Blocks World puzzle extracted solutions. The simulator manages the state of all blocks across stacks while enforcing the puzzle's movement constraints. Each move is executed in the puzzle setup with three-layer validation: verifying stack indices are within bounds, confirming the source stack contains blocks, and ensuring the specified block is at the top of its stack (enforcing the top-block-only movement rule). Upon successful validation, the block transfer is executed and the block is popped from the source stack and appended to the destination stack. Finally, the complete solution sequences of block movements are processed and verified that the resulting configuration matches the target goal state.

### A.3  Implementation Details

**Configurations.** Our experiments primarily utilized reasoning models and their corresponding non-reasoning counterparts to enable thorough analysis of the RL-enabled long CoT (i.e., thinking process). We specifically selected Claude 3.7 Sonnet (thinking/non-thinking) and DeepSeek-R1/V3 due to their ability to provide access to intermediate reasoning traces (thinking tokens), a critical requirement for our analysis. For experiments focused solely on final accuracy metrics, we also included results from OpenAI's o3-mini models. For Claude 3.7 Sonnet (w. and w/o extended thinking) models we used maximum generation budget of 64,000 tokens, accessed through the API interface. Temperature is the default 1.0 for all API rus (Claude-3.7-Sonnet and o3-mini runs). The experiments with DeepSeek-R1, DeepSeek-V3, and DeepSeek-R1-Distill-Qwen-32B are conducted on local servers with maximum generation length set to 64,000 and temperature set to 1.0. For each puzzle instance and complexity level, the results are reported on 25 samples per model. We apply a filtering process to ensure all analyzed samples are following the requested response format, including move sequences and reasoning steps as specified.

**Solution Extraction.** A custom extraction pipeline was developed to process model responses and intermediate reasoning traces (thoughts). The pipeline consists of several key components. We implemented a flexible regex-based extractors to identify potential solution attempts in both the final response and thinking trace. The extraction process identify solution patterns using regular expressions (both explicit "`moves =`" patterns and alternative bracket-based solutions). We process and clean each extracted candidate solution by ($i$) Removing comments from the list (text following "#" in any line), and ($ii$) Normalizing move formats to what suggested in context to ensure consistent structure. Then, we validate solution format and structure to filter out invalid matches. During the extraction, we also capture metadata of token position for each extracted solution. Notably, for accurate position tracking within thinking traces, we employed the same tokenizer (`cl100k_base`) as the corresponding model to count tokens across all experiments. Token positions were also normalized with respect to thought length to enable cross-sample comparison. Finally, we make sure

that the recorded solutions within the thought trace are unique and duplicate solutions (identical move sequences) were filtered. In case of duplicate solutions, only the first solution is recorded for analysis.

**Solution Evaluation.** After extraction, each solution candidate is passed to the corresponding rigorous puzzle simulator for fine-grained verification and failure analysis. The simulator takes a solution as list of moves and evaluate that with respect to the puzzle (check App. A.2 for details of each puzzle simulator). Each move in the compositional solution is executed sequentially according to previous moves and the puzzle rules. At the end, the final state obtained from all moves in the sequence is compared to the goal state of puzzle to determine full solution correctness. For incorrect solutions, details of first failure move and the type of failure is also collected during the move verification with puzzle simulator.

**Execution of Prescribed Steps.** In addition to open-ended problem solving across different puzzles, we also conducted focused experiments to test how providing the explicit solving algorithm guidance with prescribed steps would affect behavior of these reasoning models (Sec. 4.4).

We expected that finding and devising solution from scratch should require substantially more computation for model (e.g., for search and verification) than just following a given algorithm's steps. However, results over two puzzles (Tower of Hanoi and Checker Jumping) in Figures 8 show that reasoning models' behavior does not change that much and the collapse still occurs at roughly same points as before with this setting. This finding strengthens evidence that the limitation is not just in problem-solving and solution strategy discovery but also in consistent logical verification and step execution limitation throughout the generated reasoning chains.

For example, models are provided with a complete recursive algorithm of solving Tower of Hanoi and Checker Jumping puzzles as follows. These algorithm scratchpads were appended to the standard problem prompt to test its impact on reasoning behavior.

---

**Example of Prescribed Algorithm for Tower of Hanoi**

Here is a pseudocode of recursive algorithm to solve the puzzle:

```
ALGORITHM Solve(n, source, target, auxiliary, moves)
    // n = number of disks to move
    // source = starting peg (0, 1, or 2)
    // target = destination peg (0, 1, or 2)
    // auxiliary = the unused peg (0, 1, or 2)
    // moves = list to store the sequence of moves

    IF n equals 1 THEN
        // Get the top disk from source peg
        disk = the top disk on the source peg
        // Add the move to our list: [disk_id, source, target]
        ADD [disk, source, target] to moves
        RETURN
    END IF

    // Move n-1 disks from source to auxiliary peg
    Solve(n-1, source, auxiliary, target, moves)

    // Move the nth disk from source to target
    disk = the top disk on the source peg
    ADD [disk, source, target] to moves

    // Move n-1 disks from auxiliary to target
    Solve(n-1, auxiliary, target, source, moves)
END ALGORITHM
```

---

Note: When executing this pseudocode, track which disk is currently on top of each peg. The
disk IDs in the moves list should correspond to the actual disk being moved.
You can use this algorithm as a scratchpad to help you solve the problem step by step.

---

**Example of Prescribed Algorithm for Checker Jumping**

Here is a pseudocode of recursive algorithm to solve the puzzle:

```
ALGORITHM Solve(board, goal, moves)
    // board = current state (string with 'R', 'B', '_')
    // goal = target configuration
    // moves = list to store the sequence of moves

    IF board equals goal THEN
        RETURN moves
    END IF

    FOR each position i in board DO
        IF board[i] is 'R' THEN
            // Try moving R piece right (step or jump)
            TryMove('R', i, i+1, board, goal, moves)
            TryMove('R', i, i+2, board, goal, moves)
        END IF

        IF board[i] is 'B' THEN
            // Try moving B piece left (step or jump)
            TryMove('B', i, i-1, board, goal, moves)
            TryMove('B', i, i-2, board, goal, moves)
        END IF
    END FOR

    RETURN null // No solution found
END ALGORITHM

FUNCTION TryMove(piece, from, to, board, goal, moves)
    IF to is valid position AND board[to] is '_' THEN
        // Make the move
        new_board = copy of board
        SET new_board[from] = '_'
        SET new_board[to] = piece

        // Recursively solve
        result = Solve(new_board, goal, moves + [piece, from, to])

        RETURN result
    END IF
END FUNCTION
```

Note: When executing this pseudocode, use backtracking and use TryMove function to validate
that moves follow the rules (R moves right, B moves left, jumps are over opposite colors).
You can use this algorithm as a scratchpad to help you solve the problem step by step.

## A.4    Details on Computational Complexity

### A.4.1    Compositional Depth Characterization

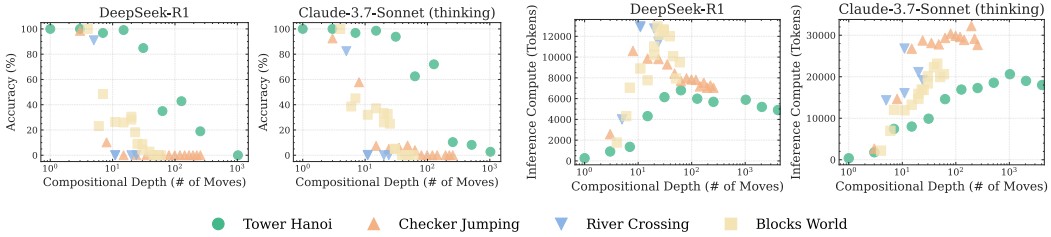

Figure 11: Inference compute (tokens) and accuracy versus compositional depth (number of required moves) across puzzle environments for DeepSeek-R1 and Claude-3.7-Sonnet (thinking).

Compositional depth is the number of sequential operations (i.e., moves) required to completely solve the puzzle. Figure 10 demonstrates how this depth scales with problem size ($N$) across our four puzzle environments. Each puzzle has a distinct growth pattern, reflecting its underlying computational complexity. For example, Tower of Hanoi shows exponential growth ($2^N - 1$), and Checker Jumping displays quadratic scaling ($(N + 1)^2 - 1$). The River Crossing and Blocks World puzzles show more moderate, near-linear growth with $N$. These varying compositional depth profiles enable us to better evaluate how reasoning models approach different types of sequential reasoning challenges and if their accuracy is always correlated with the computational complexity required to solve the puzzle. More details regarding this analysis is provided in Figure 11.

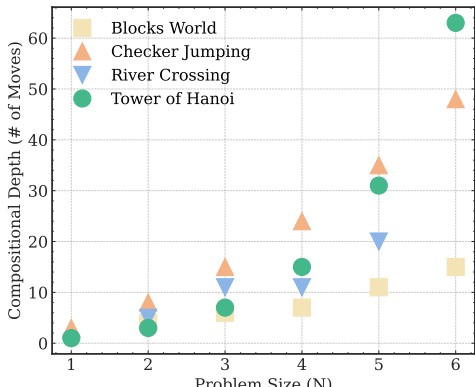

Figure 10: Compositional depth (number of moves required) across different problem sizes for our four puzzle environments.

### A.4.2 Performance vs Compositional Depth

While intuition suggests a negative correlation between problem computational complexity and model accuracy, our analysis reveals a more nuanced relationship between compositional depth and LRM performance which needs deeper discussion. Figure 11 demonstrates this across three state-of-the-art reasoning models (Claude-3.7-Sonnet w. thinking, DeepSeek-R1, and o3-mini) on our puzzle suite. Within individual puzzle types, we observe the expected negative correlation: as compositional depth increases, model accuracy consistently decreases. However, across different puzzle types, this relation does not hold. Models may struggle with puzzles of lower compositional depth while succeeding on different puzzles with higher compositional depth. For instance, models achieve >50% accuracy on Tower of Hanoi instances requiring approximately $\sim 10^2$ moves, yet consistently fail on River Crossing or Cross Jumping puzzles with substantially lower compositional depth ($\sim 10^1$ moves).

Although River Crossing may generally have a higher branching factor than Tower of Hanoi, this comparison of computational complexity is asymptotic and does not hold for the small N values used in our experiments where collapse occurs. The search space for a valid 11-move (N=3) River Crossing solution is vastly smaller than the search space for a 255-move (N=8) Tower of Hanoi solution where models start to struggle. This might reflect the limited availability of River Crossing examples with larger N values in web-based training data, meaning LRMs encountered fewer such instances during their training phase. Ultimately, LLMs and LRMs represent complex systems whose problem-solving capabilities cannot be predicted solely from computational complexity without considering their training data exposure. Their approach to handling complexity aligns more closely with the solution patterns they learned during training rather than the actual computational complexity of the problems themselves. That's why our focus on complexity is mostly to track model behavior within each puzzle setting rather than between the puzzles.

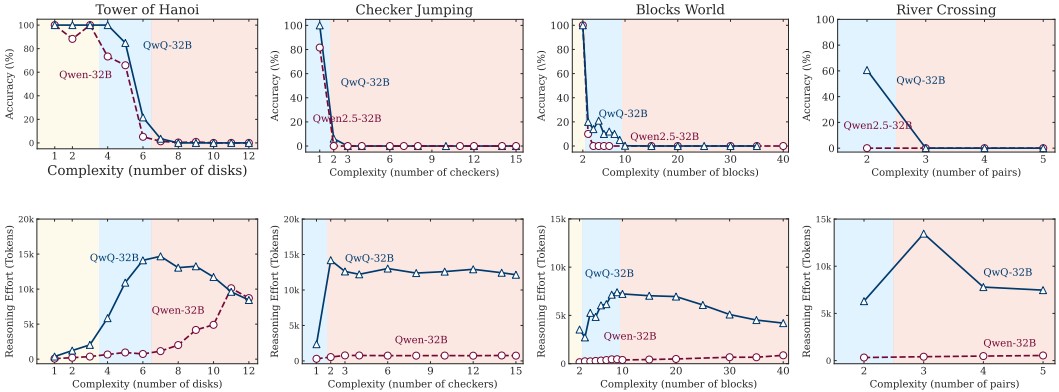

Figure 12: Comparison of reasoning ("thinking") and non-reasoning ("non-thinking") models for the *QwQ-32B* and *Qwen2.5-32B* backbone pair across all the controlled puzzle environments. **Top:** task accuracy (%) as a function of problem complexity. **Bottom:** reasoning effort (measured by the number of thinking tokens) versus problem complexity. Shaded regions correspond to the three observed regimes of reasoning behavior— low-complexity , mid-complexity , and high-complexity . Similar to other model pairs (DeepSeek-R1 vs V3 and Claude-3.7-Sonnet w./w.o. thinking), QwQ-32B demonstrates an initial advantage in the mid-complexity regime but ultimately exhibits collapse behavior at higher complexity levels with the counterintuitive decrease of reasoning effort after the collapse.

## A.5 Inference Compute vs Compositional Depth

We extend our analysis by examining how inference compute (i.e., the number of inference generated tokens) scales with compositional depth—a proxy for the number of reasoning steps/moves required to reach a solution. Figure 11 presents this relation alongside task accuracy for the DeepSeek-R1 and Claude-3.7-Sonnet (thinking) models across the four puzzle environments. Interestingly, compute allocation across puzzles is not monotonic with respect to compositional depth. We also observe inconsistent compute allocation across puzzles, with models sometimes spending more tokens on problems with lower compositional depth and vice versa—further supporting that their approach to complexity doesn't necessarily align with the actual computational complexity of the problem.

## A.6 Extended Results and Analysis

**Additional Model Pairs (QwQ-32B vs Qwen2.5-32B).** To further validate the generality of our findings, we extended our experiments to include the Qwen backbone family, comparing the reinforcement-learning–enhanced reasoning/thinking model QwQ-32B with its standard non-reasoning/non-thinking counterpart Qwen2.5-32B. These models follow the same experimental setup as in the main paper, evaluated across the four controlled puzzle environments: Tower of Hanoi, Checker Jumping, Blocks World, and River Crossing. As shown in Figure 12, this additional model pair exhibits similar trends with those observed for the DeepSeek-R1 vs V3 and Claude-3.7-Sonnet (w./w.o. thinking) pairs. Specifically, three distinct regimes of reasoning behavior emerge with respect to problem complexity. Quantitatively, the collapse points occur around N = 7 for Tower of Hanoi, N = 2 for Checker Jumping, N = 10 for Blocks World, and N = 3 for River Crossing. QwQ-32B also shows a counterintuitive reduction in reasoning effort (token count) near and beyond the collapse point, suggesting the similar scaling limit behavior with respect to complexity.

**Eliminating Sampling Effects.** To examine whether sampling causes collapse behavior, we conducted ablation experiments using temperature zero to eliminate sampling effects on solution length. These experiments used the DeepSeek-R1 reasoning model on local servers to ensure precise temperature control. The results demonstrate that removing sampling does not prevent collapse—we observe failure at roughly the same points across all puzzles, including Tower of Hanoi. This indicates that sampling of long sequences is not the primary cause of the observed collapse behavior. In fact, sampling actually helps delay collapse slightly in Tower of Hanoi and Blocks World puzzles, as

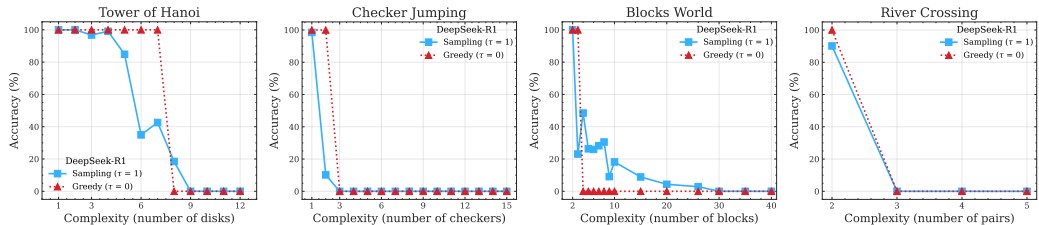

Figure 13: Performance comparison of the DeepSeek-R1 model with sampling versus greedy generations (temperature=0) across all puzzles.

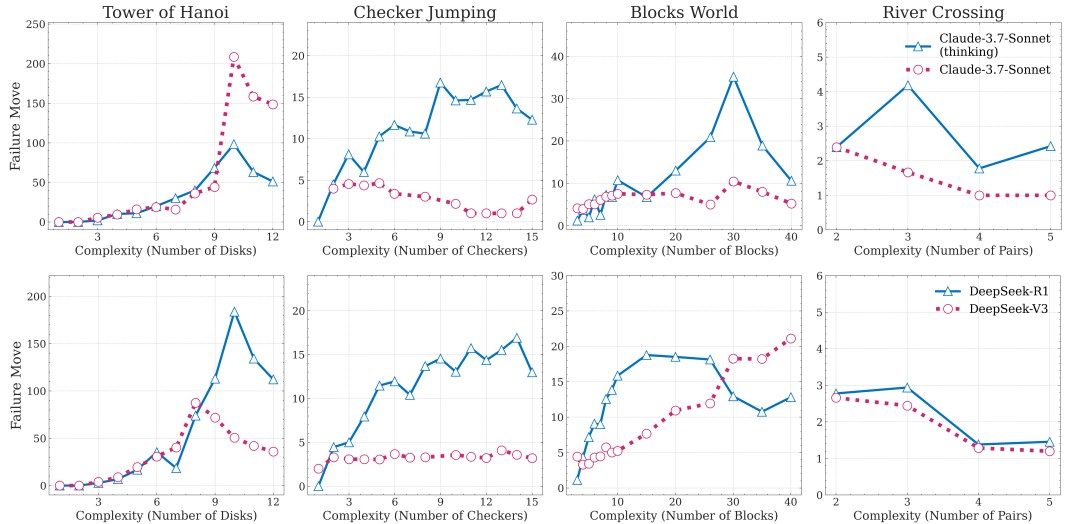

Figure 14: The first failure move versus problem complexity ($N$) comparison for thinking and non-thinking models across puzzle environments. **Top**: Claude-3.7-Sonnet comparison; **Bottom**: DeepSeek-R1 vs DeepSeek-V3.

shown in Figure 13. For instance, in Tower of Hanoi, the R1 model with temperature 0 collapses at N=8, whereas with sampling enabled, it maintains 18.2% accuracy at N=8 and delays collapse until N=9. In Blocks World also the R1 model with temperature 0 collapses at N=4, whereas with sampling enabled, it maintains 44.1% accuracy at N=4 and delays collapse until N=30.

**Failure Analysis.** Understanding where models fail within the compositional reasoning steps provides insights beyond binary success metrics. Our accuracy evaluation requires perfect execution of entire move sequences—a single incorrect move results in failure. To examine failure patterns more granularly, we analyze the compositional depth at which models first make incorrect moves across varying problem complexity levels.

Figures 14 and 15 show the failure move ID versus problem complexity ($N$) within the solution sequence across different models. In Figure 14, the top row compares Claude-3.7-Sonnet with and without extended thinking capabilities, while the bottom row compares deepseek model variants: DeepSeek-R1 (thinking) with DeepSeek-V3 (non-thinking). Figure 15 also shows this for the o3-mini model variants. These comparisons demonstrates how thinking mechanisms of LRMs influence failure patterns in compositional reasoning tasks of puzzles with respect to complexity. Our analysis reveals several interesting findings. First, we observe that the failure move usually happens much earlier than the final move required to solve the puzzle across all environments. For example, the Tower of Hanoi with N=10 requires $\sim 10^3$ moves, but the first failure move typically occurs $\sim 10^2$ moves (approximately 10% of the solution length), or N=8 requires 255 moves to complete, but the first failure move typically occurs around move 50 (approximately 20% of the solution length). Similarly, the River Crossing puzzle with N=3 requires 11 moves to solve, but the first failure move happens as early as move 4. Second, models exhibit inconsistent and non-monotonic failure behavior with respect to problem complexity—instances where models fail earlier in the solution sequence

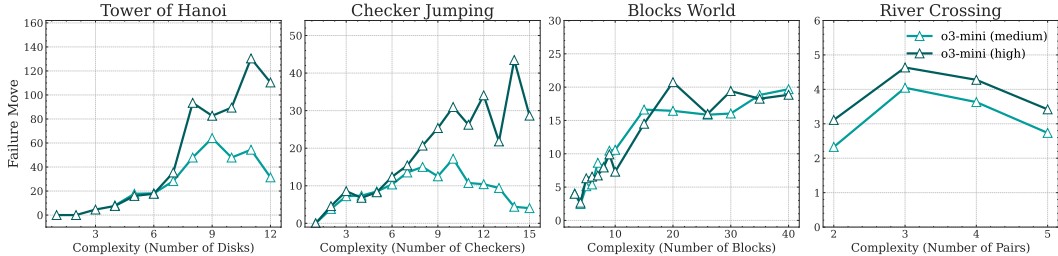

Figure 15: The first failure move versus problem complexity ($N$) comparison for o3-mini model variants across puzzle environments.

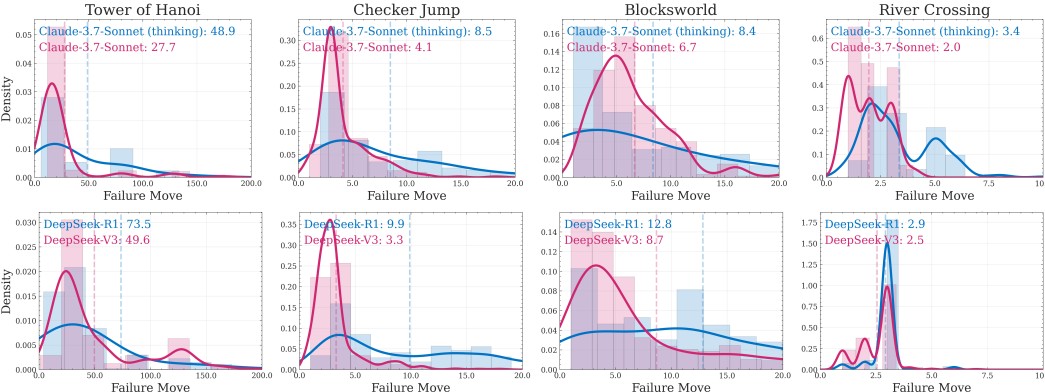

Figure 16: Density distribution of first failure moves for thinking and non-thinking models across puzzle environments. **Top**: Claude-3.7-Sonnet comparison; **Bottom**: DeepSeek-R1 vs DeepSeek-V3.

for higher $N$ values despite requiring longer overall solutions. For example, in Tower of Hanoi, models sometimes fail at below 50 moves for $N = 12$ but succeed through more than 100 moves for $N = 10$, contradicting the expectation that effective algorithmic planning and execution for the same puzzle should maintain consistent failure patterns relative to solution progress. This suggests fundamental inconsistencies in how models (both LRMs and their non-thinking standard LLM counterparts) apply learned solution strategies across different problem scales. Also, we observe that in the high-complexity regimes where both model variants experience complete accuracy collapse, e.g., Tower of Hanoi with $N \geq 8$ and Blocks World with $N \geq 30$, non-thinking models occasionally sustain performance deeper into the solution sequence and are able to fail at later moves than thinking-enabled variants. This is interesting as it shows that compositional reasoning failures in LLMs are not simply due to insufficient context length or inference compute, but rather reflect fundamental limitations in how models apply maintain and apply algorithmic consistency across problem scales.

We also analyze the distributional characteristics of failure moves to understand the consistency and reliability of model reasoning. Figure 16 presents the density distributions of failure move positions aggregated across all problem complexities for each puzzle environment, comparing thinking and non-thinking models within the same family. Based on the figure, thinking models (Claude-3.7-Sonnet with thinking and DeepSeek-R1) consistently show higher mean failure positions across all puzzles, as indicated by the dashed vertical lines showing mean of first failure in sequence of moves. However, the distribution shape of thinking models mostly show higher variance in the failure patterns. This suggests that while these models can reach deeper into solution sequences on average, their reasoning processes seem to be more instable and prone to inconsistent performance.

**Reasoning Effort Dynamics.** Figure 18 demonstrates the reasoning effort (measured by inference thinking tokens) versus problem complexity across our puzzle environments. Green dots indicate correct solutions, red crosses show incorrect ones, and blue lines track average thinking token usage at each complexity level ($N$) across different puzzles and LRMs. We observe a common pattern across most experiments with three reasoning models (DeepSeek-R1, Claude-3.7-Sonnet-thinking, o3-mini) where thinking token usage, i.e. reasoning effort, initially scales with problem complexity

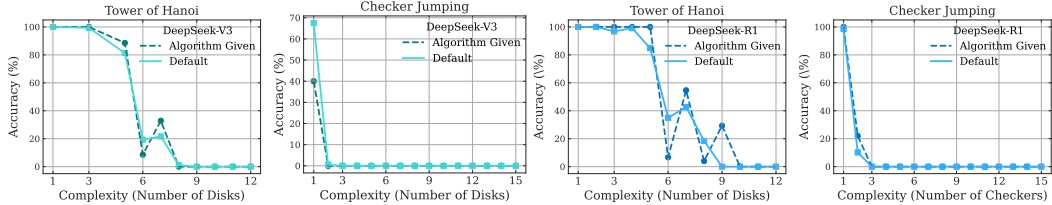

Figure 17: Effect of algorithm provision on both reasoning (thinking) and non-reasoning (non-thinking) models (*DeepSeek R1 vs V3*) across the Tower of Hanoi and Checker Jumping puzzles. "Default" denotes the original task prompt, while "Algorithm Given" includes the explicit step-by-step solving procedure in the input.

but later counterintuitively declines after reaching a model-specific complexity threshold. This suggests an interesting and fundamental scaling limit in LRM thinking process for reasoning where beyond certain complexity thresholds, models not only fail to solve problems but counterintuitively reduce their inference compute despite facing more difficult problems and being well below the context and generation limits.

**Algorithm Provision to Non-Reasoning Model.** In addition to the algorithm-provision experiments on reasoning models in Figure 8, we tested whether providing explicit solving algorithms could change collapse behavior in non-reasoning models, given recent evidence that RL-induced reasoning in LRMs may impair their instruction-following abilities [50]. As shown in Figure 17, both reasoning (DeepSeek-R1) and non-reasoning (DeepSeek-V3) models demonstrate unchanged collapse points despite modest accuracy gains from algorithm provision at certain complexity levels. This indicates that while explicit instructions might improve local execution in some cases, the collapse most likely arises from deeper limitations in compositional representation and planning rather than the instruction-following deficiencies.

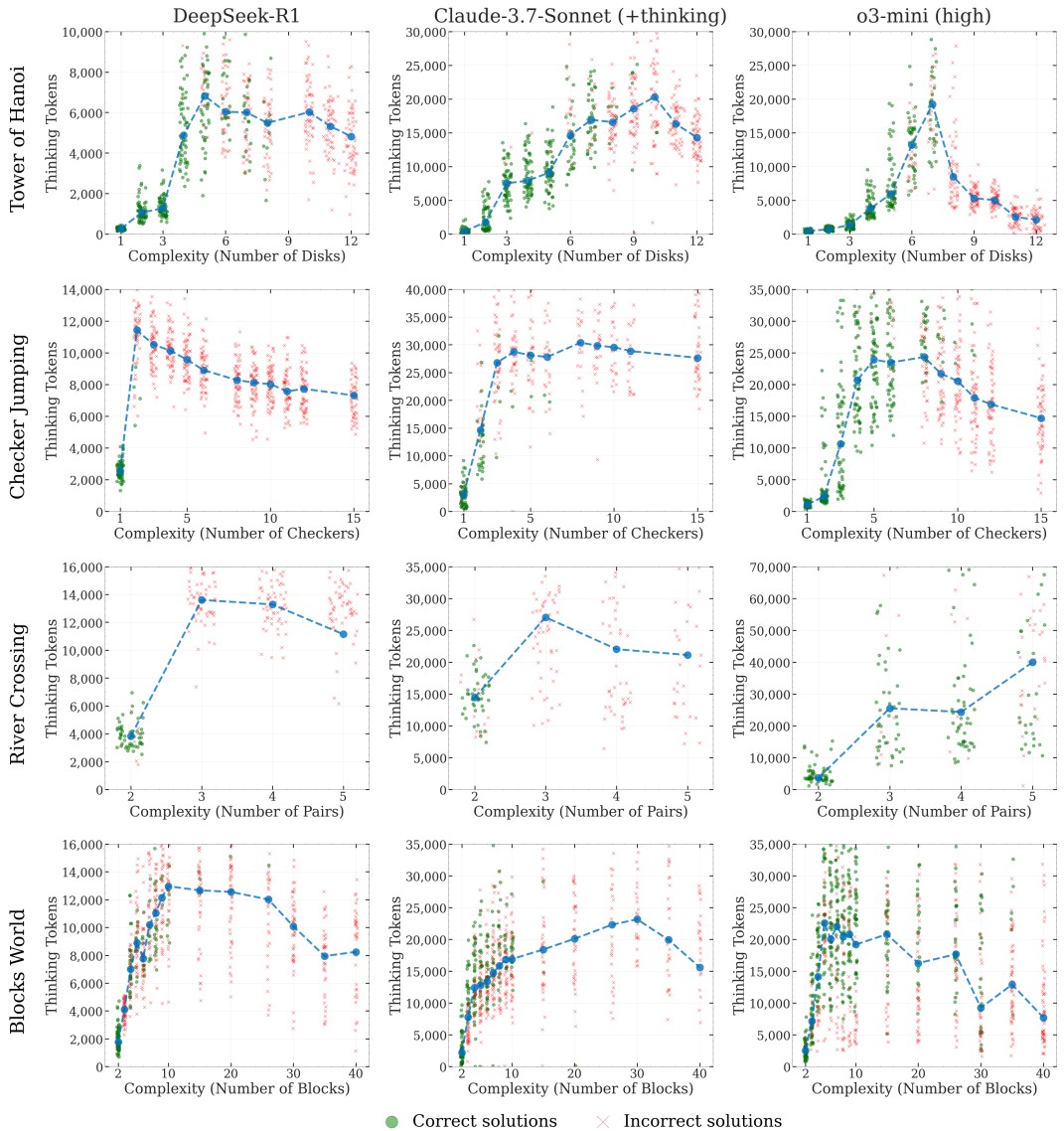

Figure 18: Detailed results on reasoning effort (measured in inference thinking tokens) versus problem complexity (N) for three LRMs (DeepSeek-R1, Claude-3.7-Sonnet with thinking, and o3-mini) across four puzzle environments.

**Qualitative Examples of Thoughts in LRMs.**   Here are some qualitative examples of thinking traces for the Claude-3.7-Sonnet (thinking) reasoning model.

Example of failures in constraint understanding and verification for River Crossing with $N = 2$:

<think>

Comprehension

Ok, this is a constraint-based puzzle. Let me try to understand the puzzle. We have two actors: a1 and a2, two agents: A1 and A2 where ... The constraints are: (1) no actor can be in the presence of another agent unless their own agent is also present, (2) the boat cannot travel empty, ... Let's understand the constraints more clearly .... Wait, I'm realizing that the constraint is not clear to me. Let me restate it.. Actor a1 cannot be in the presence of agent A2 unless agent A1 is also present.

...

Given these constraints, let's think through the solution. ...

---

Solution

Now let's start solving. First move: Two people have to cross from left to right. Let's try with Move 1: [a1, a2] cross from left to right Left side: A1, A2 Right side: a1, a2.

...

False violation alarm! (Both agents are still in the left bank)

Hmm, but now we have a problem. We need to send someone back from the right to the left, but neither actor can go back alone (because they'd meet the other agent without their own).

...

Let's try another approach with Move 1: [A1, A2] cross from left to right, Left side: a1, a2, Right side: A1, A2, Now, one agent has to come back so Move 2: [A1] comes back from right to left, Left side: a1, a2, A1, Right side: A2. **But now we have a problem on the left side: a2 is with A1 but without A2, which violates our constraint.**.

...

Let's try another approach with Move 1: [a1, A1] cross from left to right, Left side: a2, A2, Right side: a1, A1. Move 2: [A1] comes back from right to left Left side: a2, A2, A1 Right side: a1.

...

False violation alarm! (Agent A2 is still in the left bank)

But now a2 is with A1 without A2, which violates our constraint.

...

Let's try another approach with Move 1: [a2, A2] cross from left to right, Left side: a1, A1, Right side: a2, A2. Move 2: [A2] comes back from right to left, Left side: a1, A1, A2, Right side: a2

...

False violation alarm! (Agent A1 is still in the left bank)

But now a1 is with A2 without A1, which violates our constraint.

...

Let me try to cross one actor-agent pair at a time. Initial state: a1: L, a2: L, A1: L, A2: L, boat on Left.;

Move 1: [a1, A1] cross from left to right. State after Move 1: a1: R, a2: L, A1: R, A2: L, boat on Right;

Move 2: [A1] comes back from right to left State after Move 2: a1: R, a2: L, A1: L, A2: L, boat on Left;

**Move 3: [a2, A2] cross from left to right. State after Move 3: a1: R, a2: R, A1: L, A2: R, boat on Right;**

Move 4: [A2] comes back from right to left. State after Move 4: a1: R, a2: R, A1: L, A2: L, boat on Left;

Move 5: [A1, A2] cross from left to right. State after Move 5: a1: R, a2: R, A1: R, A2: R, boat on Right.

...

Violation missed in Move 3! (A2 and a1 are in the left bank without A1).

</think>

Perfect! Everyone is now on the right side, and no actor was ever in the presence of another agent without their own agent also being present.
...
So this solution correct. All right, I'll make this solution my final answer:
moves=[['a1', 'A1'], ['A1'], ['a2', 'A2'], ['A2'], ['A1', 'A2']]
</think>

---

Final Answer

Final solution for the sequence of boat moves is:
" moves=[['a1', 'A1'], ['A1'], ['a2', 'A2'], ['A2'], ['A1', 'A2']] "
...

Wrong conclusion! The final solution is incorrect.

This solution fulfills all requirements - everyone starts on the left side and ends on the right side, the boat always carries 1 or 2 people, and no actor is ever in the presence of another agent without their own agent also being present.

Example of overthinking in the thinking traces of model for Tower of Hanoi with $N = 2$

<think>

Comprehension

Ok, let me try to understand this problem. I'm given a Tower of Hanoi puzzle with 2 disks. This is a classic problem with a well-known solution algorithm. Initial configuration:
Peg 0: [2, 1] (bottom to top) Peg 1: [] Peg 2: []
Goal configuration:
Peg 0: [] Peg 1: [] Peg 2: [2, 1] (bottom to top)
For Tower of Hanoi with $n$ disks, the minimum number of moves is $2^n - 1$. So with 2 disks, we need 3 moves.

---

Solution

Let me solve this step by step:
First move: Move disk 1 from peg 0 to peg 1.
Peg 0: [2] Peg 1: [1] Peg 2: []
Second move: Move disk 2 from peg 0 to peg 2.
Peg 0: [] Peg 1: [1] Peg 2: [2]
Third move: Move disk 1 from peg 1 to peg 2.
Peg 0: [] Peg 1: [] Peg 2: [2, 1]
**So the solution is: moves = [[1, 0, 1], [2, 0, 2], [1, 1, 2]]** This means:
Move disk 1 from peg 0 to peg 1; Move disk 2 from peg 0 to peg 2; Move disk 1 from peg 1 to peg 2;
...

Repeating the same solution (#1)

Let me double-check this solution: Initial state:
Peg 0: [2, 1] Peg 1: [] Peg 2: []
After move 1 - Move disk 1 from peg 0 to peg 1:
Peg 0: [2] Peg 1: [1] Peg 2: []
After move 2 - Move disk 2 from peg 0 to peg 2:
Peg 0: [] Peg 1: [1] Peg 2: [2]
After move 3 - Move disk 1 from peg 1 to peg 2:
Peg 0: [] Peg 1: [] Peg 2: [2, 1]
Final state:
Peg 0: [] Peg 1: [] Peg 2: [2, 1]
This matches our goal configuration, so our solution is correct.
..

Repeating the same solution (#2)

Double checking:

Disk 1 (smallest) on top of Disk 2 (largest) on Peg 2. This looks correct, but let's look more closely at the specified goal configuration.

Goal configuration: Peg 0: (empty) Peg 1: (empty) Peg 2: [ (bottom) 1 (top) There seems to be a typo or incomplete information in the goal configuration description. Let me assume that the goal is to move all disks to Peg 2, with the smallest disk (1) on top of the largest disk (2), which is the standard Tower of Hanoi goal. This would match my solution above. But if I read it literally, it says: Peg 2: [ (bottom) 1 (top) This could be interpreted as only having disk 1 on peg 2, which doesn't account for disk 2. Since this doesn't make sense for Tower of Hanoi (we need to move all disks), I'll assume the goal is to have both disks on peg 2, with disk 2 at the bottom and disk 1 on top, which matches my solution.

**So the answer is: moves = [[1, 0, 1], [2, 0, 2], [1, 1, 2]]**

...

Actually, let me carefully review the problem statement once more to make sure I understand it correctly. The goal is to: Move the entire stack to the third peg (peg 2). Follow the Tower of Hanoi rules (no larger disk on smaller disk).

This confirms that we want both disks on peg 2, with disk 2 at the bottom and disk 1 on top. So my solution is correct. Let me follow the step-by-step instructions once more:

Move disk 1 from peg 0 to peg 1:
Peg 0: [2] Peg 1: [1] Peg 2: []
Move disk 2 from peg 0 to peg 2:
Peg 0: [] Peg 1: [1] Peg 2: [2]
Move disk 1 from peg 1 to peg 2:
Peg 0: [] Peg 1: [] Peg 2: [2, 1]

**So the sequence of moves is: moves = [[1, 0, 1], [2, 0, 2], [1, 1, 2]]**

...

Actually, I just noticed an ambiguity. The problem statement says the goal configuration has: Peg 2: [ (bottom) 1 (top) This could be interpreted as having disk 1 (the smallest) as both the bottom and top disk on peg 2, which would mean only disk 1 is on peg 2, and disk 2 is somewhere else. But this can't be the case because the problem statement also mentions "move the entire stack to the third peg", which means all disks should be on peg 2. Let's assume the goal configuration description meant to say: Peg 2: [ 2 (bottom) 1 (top)] This would be consistent with the standard Tower of Hanoi goal. Let me ensure I haven't misunderstood the problem. The problem statement says the goal is to move the entire stack to the third peg. This means all disks should be on peg 2 at the end. Following the Tower of Hanoi rules, the disks should be arranged in ascending order from bottom to top, which means: Peg 2: [2, 1] (bottom to top)

**This is consistent with my solution: moves = [[1, 0, 1], [2, 0, 2], [1, 1, 2]]**

...

Actually, I'm second-guessing myself because the problem mentioned "find the minimum sequence of moves." For Tower of Hanoi with 2 disks, the minimum number of moves is indeed 3, which matches our solution.

Let's be sure we've interpreted the notation correctly: "[disk id, from peg id, to peg id]" - disk id refers to the disk number (1 for the smallest, 2 for the next, and so on). - from peg refers to the peg id from which the disk is being moved (0, 1, or 2). - to peg refers to the peg id to which the disk is being moved (0, 1, or 2).

In our solution: 1. Move Disk 1 from Peg 0 to Peg 1: [1, 0, 1] 2. Move Disk 2 from Peg 0 to Peg 2: [2, 0, 2] 3. Move Disk 1 from Peg 1 to Peg 2: [1, 1, 2]

So the final solution is:
moves = [[1, 0, 1], [2, 0, 2], [1, 1, 2]]
I'm reassured that this is correct.

...

Actually, I'm overthinking this. Let me just follow the standard algorithm for Tower of Hanoi.
For n=2 disks, the minimum number of moves is 2**2 - 1 = 3 moves. The algorithm to move n disks from source to target using auxiliary:
Move n-1 disks from source to auxiliary. Move the largest disk from source to target. Move n-1 disks from auxiliary to target.
Applying this to our problem with n=2:
Move disk 1 from peg 0 to peg 1; Move disk 2 from peg 0 to peg 2; Move disk 1 from peg 1 to peg 2;
I'm confident this is the correct solution. This gives us: moves = [[1, 0, 1], [2, 0, 2], [1, 1, 2]]
**So in summary, the answer is: moves = [[1, 0, 1], [2, 0, 2], [1, 1, 2]]**
</think>

