# OpenReview forum: "The Illusion of Thinking: Understanding the Strengths and Limitations of Reasoning Models via the Lens of Problem Complexity"
_NeurIPS.cc/2025/Conference — NeurIPS 2025 poster_

### Official Review · Reviewer_K8Ta · 2025-06-21

**Clarity:** 3
**Significance:** 2
**Originality:** 3
**Rating:** 4
**Confidence:** 4

**Summary:**

This paper compares and contrasts reasoning models with "classic" large language models in a controlled environment, namely, solving puzzles. The puzzles have well-defined complexities to evaluate how well models perform in terms of different problem complexities. The study finds that the large reasoning models do not always show a performance gain across different complexities compared to traditional large language models: they perform comparably, or even underperform traditional models in low-complexity scenarios; outperform traditional models in medium-complexity scenarios; and collapse as traditional models do in high-complexity scenarios.

**Questions:**

Please see weaknesses above. Beyond the weaknesses, can the authors also clarify why the easy-medium-hard regimes in Figure 4 seem to vary by model and tasks?

**Ethical Concerns:**

["NO or VERY MINOR ethics concerns only"]

**Final Justification:**

My concerns have been resolved.

**Limitations:**

yes

**Paper Formatting Concerns:**

I have no formatting concerns.

**Quality:**

2

**Strengths And Weaknesses:**

Strengths:

1. The paper is well written and easy to follow (clarity). Figures present the results very well.

2. The topic is important: whether reasoning models can really "reason" or is it a misnomer? It is also important to find why and when we need reasoning models (significance).

3. The experiment is carefully designed with several tasks (quality).

4. The testing of reasoning models on complexity-controlled tasks is novel (novelty).

Weaknesses:

1. The motivation to mitigate contamination is not very well supported by the experiment setting (quality). I understand that the authors want to set up an environment that is generated on the fly, but as the authors note on page 9, line 295, that LRMs might still memorize instances during training for known tasks with specific variables. This is also well documented in existing works such as [1].

2. Can the authors estimate and clarify whether the hard problems are actually solvable within the context window of these models? The authors note that the models' output lengths for hard problems are even lower than simpler problems, but is it possible that the models know the problems are not solvable, so that they just use a lower output budget to reduce the cost and terminate the reasoning? If so, this is not a "bad" phenomenon from my point of view, as can also be seen in [2, 3] and other works to improve inference efficiency.

3. Personally, I think it is not surprising that LRMs perform comparably or underperform LLMs in low-complexity settings, outperform them in medium-complexity settings, and collapse with them in high-complexity settings. If tasks are very easy, we do not need very advanced models (e.g., what is the answer to 1+1). When tasks are very hard, even very advanced models cannot solve the problem (e.g., prove the Riemann Hypothesis). The places where LRMs are useful are exactly in the problems that are neither extremely easy nor extremely hard.

4. The significance of the tasks is questionable in my opinion. I wonder how often real-life users use reasoning models to solve logic puzzles like those examined in this paper. It is, of course, not difficult to come up with very difficult questions to make models fail, but what is important is to identify real-life critical tasks that the models still cannot do for users. This is also acknowledged in the limitations section of the paper.

[1] Prabhakar, A., Griffiths, T. L., & McCoy, R. T. (2024). Deciphering the factors influencing the efficacy of chain-of-thought: Probability, memorization, and noisy reasoning. arXiv preprint arXiv:2407.01687.

[2] Damani, M., Shenfeld, I., Peng, A., Bobu, A., & Andreas, J. (2024). Learning how hard to think: Input-adaptive allocation of lm computation. arXiv preprint arXiv:2410.04707.

[3] Manvi, R., Singh, A., & Ermon, S. (2024). Adaptive inference-time compute: Llms can predict if they can do better, even mid-generation. arXiv preprint arXiv:2410.02725.

---

> ### Author Rebuttal · Authors · 2025-07-30
>
> Thank you for dedicating your time and expertise to review our submission. Please find our responses below.
>
>
> > Can the authors estimate and clarify **whether the hard problems are actually solvable within the context window** of these models? The authors note that the models' output lengths for hard problems are even lower than simpler problems, but **is it possible that the models know the problems are not solvable, so that they just use a lower output budget** to reduce the cost and terminate the reasoning? If so, this is not a "bad" phenomenon from my point of view, as can also be seen in [2, 3] and other works to improve inference efficiency.
>
> That's a great question and thank you for raising this interesting point for discussion.
> First of all, we do clarify that based on our empirical evidence, collapse behavior observed in experiments occur within the output limits in all puzzles.
> For the Tower of Hanoi specifically, with its exponential growth which has raised questions of context-limit failures, reasoning models begin to collapse at N=7-8, corresponding to 127-255 moves (as shown in Figure 4 and Figure 10 in appendix) which is solvable well below the context window limits. To account for context limit concerns for large values of N in ToH, we have excluded N>12 from our experiments (to be updated in the camera-ready). For N<=12, the problems are solvable within the context window, yet models still lower their reasoning length (tokens) with complexity after the collapse ($\sim$N=8). This indicates that lowering reasoning budget observed in our experiments is not happening because of the unsovability of problems in the context window limits.
>
> Also, for other puzzles, the solutions are always solvable within the context window limits yet we still observe this counterintuitive lowering of reasoning budget after a complexity point. For River Crossing specifically, concerns about infeasibility for N>6 are not due to context limits but rather stem from the mathematical properties of the problem itself, which requires k>3 when N>5. Upon further experiments, we found that the puzzle dynamics shift considerably for N>5, where the optimal boat capacity is (k = 4) and fundamentally changes the problem structure. So, we have modified our environment on this puzzle to N <=5 (to be updated in the camera-ready). For problems with N<=5 where solutions are feasible and fit well within context limits, most models (except o3-mini-high) still show the counterintuitive drop in reasoning budget after collapse (N ≈ 3).
>
> > The motivation to mitigate contamination is not very well supported by the experiment setting (quality). I understand that the authors want to set up an environment that is generated on the fly, but as the authors note on page 9, line 295, that LRMs might still memorize instances during training for known tasks with specific variables. This is also well documented in existing works such as [1].
>
> Thank you for the insightful comment. We agree with the reviewer that contamination cannot be entirely ruled out by complexity, without knowing about the model training data. However, by systematically manipulating complexity per problem, we empirically observe degradation in performance of models—even when solution lengths are short, e.g., Checker Jumping (24 moves, N=4) and River Crossing (11 moves, N=3). This trend indicates that complexity can in fact act as a useful lens to probe model behavior, beyond what is captured in common uncontrollable benchmarks.
>
> > The significance of the tasks is questionable in my opinion. I wonder how often real-life users use reasoning models to solve logic puzzles like those examined in this paper. It is, of course, not difficult to come up with very difficult questions to make models fail, but what is important is to identify real-life critical tasks that the models still cannot do for users. This is also acknowledged in the limitations section of the paper.
>
> Yes. While logic puzzles may not reflect everyday user queries, they offer envrionments for controlled and step-by-step evaluation of core reasoning capabilities. We think that these tasks can serve as diagnostic tools and a good first step towards understanding the behavior of models under controllable settings: if models struggle with simple and step-by-step verifiable reasoning, it raises concerns about their reliability in more complex and open real-world scenarios.
>
> >  Beyond the weaknesses, can the authors also clarify why the easy-medium-hard regimes in Figure 4 seem to vary by model and tasks?
>
> The easy–medium–hard regimes in Figure 4 are defined based on the performance comparison for each reasoning vs non-reasoning model pair with respect to N at each task. A regime is labeled easy where non-reasoning model perform comparably or better, hard where both model types collapse, and medium where reasoning model show clear advantage.
>
>
> ---
> Thank you once again for the thoughtful review. We noticed that several concerns raised in the reviews echo some misunderstandings about our methodology and findings. In our response, we have addressed each point with detailed evidence and clarification, while incorporating revisions where appropriate. We hope that our rebuttal address the reviewer's concerns, and if so, they would consider updating their score. We’d be more than happy to engage in further discussions.

---

> > ### Comment · Reviewer_K8Ta · 2025-08-04
> >
> > Thank the authors for the reply. I have raised my score.

---

> > > ### Author Response · Authors · 2025-08-04
> > > **Thank you from Authors**
> > >
> > > Dear Reviewer K8Ta,
> > > Thank you once again for your time and valuable questions. We are glad that our rebuttal has resolved your concerns and appreciate the raised score.

---

### Official Review · Reviewer_rPRW · 2025-07-01

**Clarity:** 3
**Significance:** 2
**Originality:** 3
**Rating:** 4
**Confidence:** 4

**Summary:**

This work systematically investigates the reasoning capabilities of LLMs and Large Reasoning Models (LRMs) using controllable puzzle environments -- like Tower of Hanoi, Blocksworld, and River Crossing, amongst others -- that isolate problem complexity. The authors identify three performance regimes and show that beyond a certain complexity (third regime), LRMs exhibit a complete accuracy collapse and a surprising decline in reasoning effort. The authors raise important questions about the true nature of LRM reasoning and thinking abilities.

**Questions:**

* What is compositional depth (section 3.1)? How is it related to your definition of "complexity"?
* What kind of sampling strategy and temperature were used? Did the authors try to see if reducing the temperature creates a big difference?

**Ethical Concerns:**

["NO or VERY MINOR ethics concerns only"]

**Final Justification:**

The authors have identified and corrected the shortcomings of their experimental settings and have promised to update the results in their final version. In general, most of the conclusions drawn from the initial version remain the same, and some of the results are interesting for the community to reflect on and improve upon. I believe the rebuttal process has been useful for both the authors and me since I have had time to reflect upon some of the results and perceive them from the authors' point of view. That also helped resolve some of my earlier doubts, especially about analysing from a computational complexity perspective -- which, as the authors have argued -- might not always be the right way of evaluating LLMs.

**Limitations:**

Yes.

**Paper Formatting Concerns:**

No.

**Quality:**

2

**Strengths And Weaknesses:**

**Strengths**
* The paper addresses an important and timely question: evaluating the reasoning abilities of LLMs, and more importantly, LRMs.
* The chosen evaluation problems are fundamental and carefully selected to mitigate data contamination, which is a pervasive issue in many standard benchmarks.
* The experimental design includes comparisons between reasoning-enabled and non-reasoning counterparts of LLMs. This approach helps control for potential similarities in pretraining data and shared model behaviors, leading to better comparisons.


**Weaknesses**.
Despite a range of experiments, there seem to be serious flaws in the claims of the paper:

* **Seemingly a lack of thorough analysis of reasoning traces.** The authors repeatedly claim to perform a detailed analysis of reasoning traces. However, the experimental results suggest that this analysis may not have been conducted as thoroughly as stated
    - L282-287: Based on Figures 8a, 8b, the authors claim that the observed collapse happens around the same place in Tower of Hanoi despite providing the algorithm. The authors state that providing the algorithm should make it easier for the LLMs to output correct steps.  It is highly plausible — and even suggested by the authors themselves — that the ToH algorithm was included in the models’ pretraining data. In this case, the LLMs and LRMs might simply reproduce this known algorithm within their reasoning (CoT) traces. Consequently, the results presented in Figures 8a and 8b are unsurprising. Did the authors investigate the reasoning traces for any such behavior?
    - Paper [1] shows that for n>=6, k>=4 is necessary to have a solution at all in River Crossing. However, it appears that the authors use k=3 for n>=4, so there are no solutions (L 195).
    - Finally, in the third regime, have the authors analyzed whether LLMs and LRMs actually produce incorrect solutions, or if they instead *give up* beyond a certain point (which would explain their slight reduction in token usage in the hard regime)? It would be important to clarify whether such *lazy* solutions are counted as incorrect solutions in the reported metrics. Furthermore, have the authors considered whether it is even feasible to fit an explicit search process within the token limits of these models at larger problem sizes, despite LRMs having significantly higher token capacities?
* **Flawed complexity analysis.**  The paper’s analysis heavily relies on measuring LLM performance against problem complexity measured primarily by the number of disks, people, or blocks. This approach could be flawed. Problem difficulty cannot be accurately assessed by instance size alone; it is also governed by factors such as action space size (i.e., branching) and requirement for backtracking (for dead ends).
   - For e.g., River Crossing problems have a significantly larger action space than ToH. Even assuming a conservative branching factor ~ 10, a solution of length 3–4 already entails searching through $10^3 - 10^4$ possible action sequences, not accounting for backtracking necessitated by dead ends. This large search space differentiates these tasks from ToH (with either 1,2,3 actions), which has a narrow and structured move space despite a longer solution length. This discrepancy is further illustrated by Figure 6 (Accuracy and Thinking Tokens vs. Complexity). If plotted on a comparable scale along the x-axis, it becomes evident that the number of thinking tokens in River Crossing problems grows rapidly even at low “complexity” levels (e.g., with just 3 people), indicating substantial *hidden difficulty* not captured by simple size metrics.
    - Additionally, recent works have emphasized analyzing LLM and LRM performance using computational complexity frameworks. Notably, [2] demonstrates that LLMs and LRMs struggle in "hard" regions of Satisfiability problems where no effective heuristics exist, even when instance sizes are modest. There, hardness is characterized by the ratio $\alpha$ = #clauses / # variables, and it is shown that neither the number of clauses nor variables alone explains problem difficulty -- directly contradicting the assumptions made in this paper. While I acknowledge that computational complexity analyses can themselves be imperfect (e.g., due to dependencies on the model’s pretraining data), this does not justify disregarding such frameworks entirely and attributing the observations instead to *puzzling behavior* (section 4.4)

[1] River Crossing Problems: Algebraic Approach, Efimova et al., 2018
[2] Have Large Language Models Learned to Reason? A Characterization via 3-SAT Phase Transition; Hazra et al., 2025

---

> ### Author Rebuttal · Authors · 2025-07-30
>
> Thank you for dedicating your time and expertise to review our submission. Please find our responses below.
>
> > L282-287: Based on Figures 8a, 8b, the authors claim that the observed collapse happens around the same place in Tower of Hanoi despite providing the algorithm. The authors state that providing the algorithm should make it easier for the LLMs to output correct steps. It is highly plausible — and even suggested by the authors themselves — that the ToH algorithm was included in the models’ pretraining data. In this case, the LLMs and LRMs might simply reproduce this known algorithm within their reasoning (CoT) traces. Consequently, the results presented in Figures 8a and 8b are unsurprising. Did the authors investigate the reasoning traces for any such behavior?
>
> The critic suggests that the findings of algorithm execution limitation in LRMs may be only an artifact of the ToH’s well-known recursive algorithm being already memorized by models, rather than indicating a general limitation in algorithm execution. To investigate this, we conducted additional experiments on Checker Jumping puzzle, with some different characteristics: it requires fewer moves than ToH, exhibits earlier collapse behavior, and appears to have less algorithmic familiarity. If the critic’s hypothesis were correct, we should observe clear benefits from algorithm provision in this problem. However, our results show very similar patterns for both puzzles (to be included in the camera ready). Even when given explicit algorithm steps—requiring only execution rather than problem-solving—models show no meaningful improvement on either puzzle, with collapse happening at similar points. These results further validate our findings regarding limitations in the execution of logical steps and suggest a more fundamental failure mode rather than one specific to the ToH.
>
> > Paper [1] shows that for n>=6, k>=4 is necessary to have a solution at all in River Crossing. However, it appears that the authors use k=3 for n>=4, so there are no solutions (L 195).
>
> Thank you for pointing this out. Upon further experiments, we found that the puzzle dynamics shift considerably for N>5, where the optimal boat capacity is (k = 4) and fundamentally changes the problem structure. Indeed this has been an overlook from us and we have currently modified our environment to N <=5 to address this. We do plan to update the camera-ready accordingly. However, we would like to note that this modification does not affect our core findings, as performance of models mostly collapse earlier from N=3.
>
> > Finally, in the third regime, have the authors analyzed whether LLMs and LRMs actually produce incorrect solutions, or if they instead give up beyond a certain point (which would explain their slight reduction in token usage in the hard regime)? It would be important to clarify whether such lazy solutions are counted as incorrect solutions in the reported metrics. Furthermore, have the authors considered whether it is even feasible to fit an explicit search process within the token limits of these models at larger problem sizes, despite LRMs having significantly higher token capacities?
>
> Thank you for the thoughtful question.
> Regarding the feasibility of solutions in token limits, our empirical evidence demonstrates that collapse behavior observed in experiments occur within the token limits in all puzzles. For the ToH specifically, with its exponential growth raising questions of context-limit failures, reasoning models begin to collapse at N=7-8, corresponding to 127-255 moves (as shown in Figure 4 and 10 in appendix) which is well within the generation limits. More importantly, if we look deeper into the failure cases (like in Figure 8c and 11 in appendix), we see that the first failure move actually happens much sooner than the final move. For example, for ToH with N=10 (requiring 1023 moves), failure typically occurs within the first $\sim$100 moves (10% of solution length); for N=8, which requires 255 moves, failure occurs around $\sim$40 moves (15% of solution length). This indicates that model failures and particularly collapse behavior happen much earlier and are not due to premature outputs because of token limits. To account for context limit concerns for large values of N in ToH, we have excluded N>12 from experiments. We do plan to update the camera-ready accordingly.
> Notably, N>12 lies within the third region, post-collapse (N=8), so this modification does not affect our core findings for this puzzle. For all other puzzles, solution lengths are short enough that search process easily fits within context limits and reach to final move.
>
> Regarding incorrect solutions versus giving up, can reviewer clarify what they mean? we confirm that models continue generating full solutions after the first failure move until reaching end /token limits. However, near collapse, they start to counterintuitively reduce reasoning effort with complexity (less thinking and reflecting) which is observed across all puzzles regardless of the solution length.
>
>
> > Flawed complexity analysis. The paper’s analysis heavily relies on measuring LLM performance against problem complexity measured primarily by the number of disks, people, or blocks. This approach could be flawed. Problem difficulty cannot be accurately assessed by instance size alone; it is also governed by factors such as action space size (i.e., branching) and requirement for backtracking (for dead ends).
> For e.g., River Crossing problems have a significantly larger action space than ToH. Even assuming a conservative branching factor ~ 10, a solution of length 3–4 already entails searching through $10^3 - 10^4$ possible action sequences, not accounting for backtracking necessitated by dead ends. This large search space differentiates these tasks from ToH (with either 1,2,3 actions), which has a narrow and structured move space despite a longer solution length. This discrepancy is further illustrated by Figure 6 (Accuracy and Thinking Tokens vs. Complexity). If plotted on a comparable scale along the x-axis, it becomes evident that the number of thinking tokens in River Crossing problems grows rapidly even at low “complexity” levels (e.g., with just 3 people), indicating substantial hidden difficulty not captured by simple size metrics.
>
> We respectfully disagree with the reviewers' assertion that our analysis is flawed.
> First of all, the statement pointed out by reviewer is not mathematically correct. While the branching factor in River Crossing is larger than in ToH, the search space for a valid 11-move solution (N=3) is vastly smaller than the search space for a 127-move (N=7) ToH solution. Specifically, in River Crossing with N=3, search space contains of $2^{2N+1}$=128 states and in ToH with N=7 search space contains of $3^N$=2187 states, yet models fail on the former and succeed on the latter. This indicates that how LLMs approach complexity does not necessarily corresponds to the actual computational complexity of the problem and more to the learned solution distributions. That’s why our focus on complexity is mostly to track model behavior within each puzzle setting rather than between the puzzles.
>
> Second, reviewer states that we solely rely on the problem size (N) while neglecting the search space and compositional depth with N, however, we have already discussed about this compositional depth for all puzzles in Appendix A.3 (Figure 9) and show the performance with respect to it (Figure 10) which is what reviewer is suggesting as "plot with comparable x-axis". Still, we observe that even tasks with low depth can face collapse, indicating that learned solution distribution is most likely the dominant factor for model behavior.
>
> > What is compositional depth of (section 3.1)? How is it related to your definition of "complexity"?
>
> The compositional depth of all puzzles across N is already discussed in Appendix A.3.
>
> > ..While I acknowledge that computational complexity analyses can themselves be imperfect (e.g., due to dependencies on the model’s pretraining data), this does not justify disregarding such frameworks entirely..
>
> Computational complexity has been already discussed in the Appendix A.3. As model behavior does not necessarily reflect the computational complexity (elaborated above with multiple evidence), we mostly rely on empirical observations within each task, which better capture model behavior given the unknowns in training data.
>
>
> > What kind of sampling strategy and temperature were used? Did the authors try to see if reducing the temperature creates a big difference?
>
> As noted in our experimental setup and implenetation details (sec 4.1 and A.2 in appendix), currently temperature 1 is used for sampling. In response to the reviews, we conducted additional ablation experiments with temperature $\tau=0$ on open-source DeepSeek-R1 (to ensure precise local control). Our results show that reducing temperature and eliminating sampling generally does not prevent collapse—in fact, sampling often slightly delays it. For example, Tower of Hanoi with $\tau$=0 causes collapse at N=8, while sampling extends performance to N=9 (18.2% accuracy at N=8). In Blocks World also= $\tau$=0 causes collapse at N=4 versus N=30 with sampling.
> We will include this analysis in the camera-ready version.
>
> ---
> Thank you once again for the thoughtful review. We noticed that several concerns raised in the reviews echo some misunderstandings about our methodology and findings. In our response, we have addressed each point with detailed evidence and clarification, while incorporating revisions where appropriate. We hope that our rebuttal address the reviewer's concerns, and if so, they would consider updating their score. We’d be more than happy to engage in further discussions.

---

> ### Comment · Reviewer_rPRW · 2025-08-03
> **Follow-up by Reviewer rPRW**
>
> Thank you, authors, for your answers. I have some follow-up questions.
>
> ` To investigate this, we conducted additional experiments on Checker Jumping puzzle, with some different characteristics: it requires fewer moves than ToH, exhibits earlier collapse behavior, and appears to have less algorithmic familiarity. If the critic’s hypothesis were correct, we should observe clear benefits from algorithm provision in this problem. However, our results show very similar patterns for both puzzles`
>
> Thank you for running the new experiment. However, it still goes back to my main question of whether the authors "actually" analyzed the reasoning traces. Could the authors clarify whether the algorithm was generated by the models (as part of their reasoning traces) in either case? For instance, when I tried the Checker Jumping puzzle myself on ChatGPT, the first thing it did was to generate a python algorithm for the same. Now, I understand that your analysis requires no access to tools; however, could it be that the algorithm is also part of its training data and that it outputs the algorithm in its reasoning traces?
>
> `Regarding incorrect solutions versus giving up, can reviewer clarify what they mean? `
> Again, thank you for the answer and the detailed analysis. Given the nature of these puzzles and the fact that they can be solved through offline planning, it’s expected that models might backtrack when they first encounter a contradiction. Indeed, LRMs are trained to exhibit such "chaining" behaviors. My question is this: were the "final" solutions incorrect in every single case, or were there instances where the model effectively gave up (e.g., producing statements like "if you keep going like this..." or "this is taking too long...") rather than producing a fully incorrect solution? Would such outputs be classified differently?
>
> `We respectfully disagree with the reviewers' assertion that our analysis is flawed. First of all, the statement pointed out by reviewer is not mathematically correct. While the branching factor in River Crossing is larger than in ToH, the search space for a valid 11-move solution (N=3) is vastly smaller than the search space for a 127-move (N=7) ToH solution. Specifically, in River Crossing with N=3, search space contains of =128 states and in ToH with N=7 search space contains of
> =2187 states, yet models fail on the former`
>
> What’s crucial to consider here is the sequence of actions, since the models must process them sequentially. In ToH, the action process is highly structured and linear, with roughly 2 valid actions available per state. River Crossing, in contrast, has a much larger branching factor—around 10 possible actions per state. This means that after just 4 moves, there could already be $10^4$ possible action sequences. (Apologies for previously using the term “state space”; I was indeed referring to action sequences.). This explosion happens very early, which forces the model not only to backtrack whenever it hits a dead end but also to continually track and evaluate the optimal path forward. While I agree that ToH could indeed have been part of the LLMs' training data, its simple recursive solution logic likely makes it easier for the models to pick up approximate heuristics.

---

> > ### Author Response · Authors · 2025-08-04
> > **Response from Authors**
> >
> > We appreciate your continued engagement and insightful questions. Here's our response:
> >
> > > However, it still goes back to my main question of whether the authors "actually" analyzed the reasoning traces. Could the authors clarify whether the algorithm was generated by the models in either case? For instance, when I tried the Checker Jumping puzzle myself on ChatGPT, the first thing it did was to generate a python algorithm for the same. Now, I understand that your analysis requires no access to tools; however, could it be that the algorithm is also part of its training data and that it outputs the algorithm in its reasoning traces?
> >
> > Thank you for the insightful question. Yes, we have analyzed the reasoning traces and we agree with the reviewer that models might have seen the algorithm in their training data. We cannot theoretically verify this as we don't know their training data. From analyzing the model traces empirically, we can speculate the model's familiarity with the solving algorithm, and in the case of Checker Jumping we observe much less appearance of algorithmic traces than in ToH (almost none in the traces of deepseek-r1 and claude sonnet 3.7 w. thinking for the selected 25 samples without python tool access).  However given all these, by algorithm provision in the prompt, we still observe very similar behavior with the same collapse point for models which likely conclude their limitations in correctly following the given algorithm steps.
> >
> > > My question is this: were the "final" solutions incorrect in every single case, or were there instances where the model effectively gave up (e.g., producing statements like "if you keep going like this..." or "this is taking too long...") rather than producing a fully incorrect solution? Would such outputs be classified differently?
> >
> > That's a great question and thank you for raising this for discussion. We confirm that in the current modified environments, the full solution including the final move always exist in the samples used for analysis. The issue that reviewer raised was true for high values of N in ToH (returning `this is taking too long...` instead of full solution). However, in the modified ToH setting with N<12, we are able to collect 25 samples for each instance which includes the full solution. For the other puzzles, solution lengths are much shorter, and full solutions with the final move are always present in the analyzed samples, yet we still observe the counterintuitive reduction in the reasoning effort.
> >
> > > What’s crucial to consider here is the sequence of actions, since the models must process them sequentially. In ToH, the action process is highly structured and linear, with roughly 2 valid actions available per state. River Crossing, in contrast, has a much larger branching factor—around 10 possible actions per state. This means that after just 4 moves, there could already be $10^4$ possible action sequences. (Apologies for previously using the term “state space”; I was indeed referring to action sequences.). This explosion happens very early, which forces the model not only to backtrack whenever it hits a dead end but also to continually track and evaluate the optimal path forward. While I agree that ToH could indeed have been part of the LLMs' training data, its simple recursive solution logic likely makes it easier for the models to pick up approximate heuristics.
> >
> > Thank you for the clarification. We agree with the reviewer that `In ToH, the action process is highly structured and linear with roughly 2 valid actions available per state. River Crossing, in contrast, has a much larger branching factor`, and `ToH could indeed have been part of the LLMs' training data, its simple recursive solution logic likely makes it easier for the models to pick up approximate heuristics`, and in fact that's why our focus on the complexity analysis is within each puzzle instead of across puzzles
> > We also agree with the reviewer on the importance of action sequences in River Crossing. However, we believe the action space comparison should also be considered between River Crossing and ToH. Using puzzle solvers to track the actual branching factor across all states, we find that for River Crossing with N=3 (11-move solution), the total number of possible actions is 318, and in ToH with N=7 (127-move solution), the total number of possible actions is 6558. Although ToH has theoretically larger space of action sequences, models succeed there but collapse on River Crossing. This again suggests that collapse is driven less by computational complexity and more by other factors such as the limitations in following puzzle constraints and the less familiarity of the puzzle for models.
> >
> > Thank you again for your thoughtful review and for the opportunity to clarify these points.

---

> > > ### Comment · Reviewer_rPRW · 2025-08-04
> > > **Response by Reviewer rPRW**
> > >
> > > Thank you again for the detailed responses. I appreciate the authors' efforts in identifying and addressing the issues in their experimental analysis during the rebuttal. That said, I do have some concerns about the extent of the changes made, particularly given that updated PDFs are no longer permitted and the process now relies heavily on mutual trust. Still, I recognize that refining the paper through feedback is a core goal of the review process. With that in mind, I'm inclined to raise my score to a borderline accept, trusting that the authors will follow through on the proposed revisions if the paper is accepted.

---

> > > > ### Author Response · Authors · 2025-08-05
> > > > **Thank you from Authors**
> > > >
> > > > Dear Reviewer rPRW,
> > > > Thank you once again for your time and insightful questions. We are glad that our rebuttal has resolved your concerns and appreciate the raised score. We will make sure that all revisions are carefully incorporated in the camera-ready version.

---

### Official Review · Reviewer_P5aZ · 2025-07-03

**Clarity:** 3
**Significance:** 3
**Originality:** 3
**Rating:** 4
**Confidence:** 3

**Summary:**

The goal of this paper is to understand the true reasoning capabilities of Large Reasoning Models (LRMs), which are LLMs that generate "thinking" steps before outputting an answer. The paper argues that current evaluation on math/coding benchmarks is basically flawed due to possible data contamination (there is some evidence with AIME 24/25) and a lack of control over the complexity of the problem (which seems an important missing feature). To address these flaws, it proposes an evaluation framework using puzzles whose complexity can be “controlled” - in particular, four puzzles including Tower of Hanoi and Blocks World. Thus, it is able to systematically vary the puzzle complexity and study not only the final accuracy but also the accuracy/quality of the intermediate reasoning steps.


The paper is basically experimental.  By studying the four puzzles, it identifies three performance regimes wrt problem complexity: non-thinking LLMs are better in the low-complexity regime, LRMs are excellent in the medium-complexity regime, and both fail in the high-complexity regime.  In particular, all models experience a full performance collapse after a certain complexity limit. Even more surprisingly, the paper suggests that as problems approach this collapse limit, the models' reasoning effort decreases, suggesting a compute scaling limit.  The paper’s experiments also suggest inefficiencies including "overthinking" of LRMs on simple problems.

**Questions:**

- How would you answer the criticism in https://arxiv.org/html/2506.09250v1
- How would you view your results with tool use?
- How would you reconcile your finding (regarding supplying code) vs instruction-following abilities these models demonstrate on say many benchmarks?

**Ethical Concerns:**

["NO or VERY MINOR ethics concerns only"]

**Final Justification:**

The responses addressed some of my concerns.  I confirm my mild positive (revised) rating.

**Limitations:**

Yes

**Quality:**

3

**Strengths And Weaknesses:**

### Strengths
1. Good amount of novelty in using controllable puzzles to study LRMs
2. Studying the reasoning trace is interesting and offers good insights
3. The three regimes are a good empirical finding
4. The paper is well written

### Weaknesses
1. Unclear if the paper’s experimental setup is completely accurate (see https://arxiv.org/html/2506.09250v1) - in particular, issues of model exceeding output  capacity (despite knowing the solution) and models faced with problems that are essentially infeasible
2. Some lack of statistical significance analysis
3. The finding of the three regimes is not too surprising - experience with thinking models has anecdotally suggested these regimes, but to its credit, this work seems to be one of the earliest to properly study and articulate it

### Minor Typos
- Line 59: "put the emphasis" -> "putting the emphasis"
- Line 125: "model thinks"-> "the model thinks"
- Line 127: "corelation" -> "correlation"
- Line 173: "in distibution" ->"in-distribution"
- Line 221: "exists three" -> "exist three"
- Line 223: "to obtain" -> "of obtaining"

---

> ### Author Rebuttal · Authors · 2025-07-30
>
> Thank you for dedicating your time and expertise to review our submission. Please find our responses below.
>
> > * Unclear if the paper’s experimental setup is completely accurate (see https://arxiv.org/html/2506.09250v1) - in particular, issues of model exceeding output capacity (despite knowing the solution) and models faced with problems that are essentially infeasible
> > * How would you answer the criticism in https://arxiv.org/html/2506.09250v1
>
> We understand the reviewers' concern and have incorporated several modifications to address valid points while clarifying misunderstandings where appropriate:
>
> **Output Length Constraints on ToH.** Our empirical evidence demonstrates that collapse behavior observed in experiments occur within the output limits in all puzzles. For the Tower of Hanoi specifically, with its exponential growth raising questions of context-limit failures, reasoning models begin to collapse at N=7-8, corresponding to 127-255 moves (as shown in Figure 4 and Figure 10 in appendix) which is well within the context limits. More importantly, if we look deeper into the failure cases (like in Figure 8c and Figure 11 in appendix), we see that the first failure move actually happens much sooner than the final move. For example, for Tower of Hanoi with N=10 (requiring 1023 moves), failure typically occurs within the first $\sim$100 moves (10% of solution length); for N=8, which requires 255 moves, failure occurs around $\sim$40 moves (15% of solution length). This indicates that model failures and particularly collapse behavior happen much earlier and are not due to premature outputs because of generation limits. To account for context limit concerns for large values of N in ToH, we have excluded N>12 from experiments. We do plan to update the camera-ready accordingly.
> Notably, N>12 lies within the third region, post-collapse (N=8), so this modification does not affect our core findings for this puzzle.
>
> **Infeasibility on River Crossing.** Upon further experiments, we found that the puzzle dynamics shift considerably for N>5, where the optimal boat capacity is (k = 4) and fundamentally changes the problem structure. Indeed this has been an overlook from us and we have currently modified our environment to N <=5 to address this. We do plan to update the camera-ready accordingly. However, we would like to note that this modification also does not affect our core findings, as performance of models mostly collapse earlier from N=3.
>
> > How would you view your results with tool use?
>
> It’s important to note that our objective is to assess models’ reasoning processes—their ability to understand problems, explore solution spaces, and execute logical steps—rather than merely achieving correct final answers. For the puzzles in our study, algorithmic solutions are very well-established as standard implementations. Allowing tool use would tell us nothing about their capacity for problem understanding, constraint reasoning, or multi-step logical execution. Tool usage becomes valuable when algorithmic codes for solving the problem are unknown and require compositional reasoning to be developed.
>
> > Some lack of statistical significance analysis
>
> We would appreciate if the reviewer could further clarify on this. Figure 13 in the appendix already presents the full distribution of results across collected samples, different models, and complexity levels. Figures 7 and 12 in appendix also provide density distributions for more in-depth analyses of intermediate reasoning traces and failure moves, which together aim to support statistical significances of findings.
>
> > How would you reconcile your finding (regarding supplying code) vs instruction-following abilities these models demonstrate on say many benchmarks?
>
> Thank you for the thoughtful question. It is true that LRMs' failure in following algorithm steps might also partially relate to their weaker instruction-following capabilities (as shown by some recent works eg [1]). To further investigate this, we conducted additional experiments during the rebuttal period with algorithm provision to non-reasoning models (eg DeepSeek-V3 and Claude 3.7 Sonnet w/o thinking). Results (to be included in the camera-ready) show that algorithm provision improves accuracy of non-reasoning models on some N, but—like LRMs—it still does not change the collapse point with increasing complexity, suggesting that failures and collapse behavior might be due to factors beyond instruction-following.
>
> [1] Li et al., When Thinking Fails: The Pitfalls of Reasoning for Instruction-Following in LLMs, 2025
>
> > The finding of the three regimes is not too surprising - experience with thinking models has anecdotally suggested these regimes, but to its credit, this work seems to be one of the earliest to properly study and articulate it
>
> Thank you for the recognition of our contribution.
>
> ---
>
> **Minor Typos.** Thank you for noting these. We will make sure to address them in the updated vesion.
>
> ---
> Thank you once again for the thoughtful review. We noticed that several concerns raised in the reviews echo some misunderstandings about our methodology and findings. In our response, we have addressed each point with detailed evidence and clarification, while incorporating revisions where appropriate. We hope that our rebuttal address the reviewer's concerns, and if so, they would consider updating their score. We’d be more than happy to engage in further discussions.

---

> > ### Comment · Reviewer_P5aZ · 2025-08-05
> >
> > Thanks for the detailed responses.  (Thanks for pointing to the appendix figs for the statistical analysis.). I am happy to raise my score, in light of the responses.

---

> > > ### Author Response · Authors · 2025-08-05
> > > **Thank you from Authors**
> > >
> > > Dear Reviewer P5aZ,
> > > Thank you once again for your time and valuable feedback. We are glad to have addressed your concerns and appreciate the raised score.

---

### Official Review · Reviewer_3cZ8 · 2025-07-03

**Clarity:** 3
**Significance:** 2
**Originality:** 3
**Rating:** 3
**Confidence:** 3

**Summary:**

This paper provides a systematic evaluation of the reasoning ability of LLMs. In its settings of puzzle problems, the unique point about its problem setting is that its complexity can be manipulated, and reasoning traces can be explicitly analyzed. Based on the experiments, the paper shows that standard language models outperform LRMs at low complexity, thinking models excel at medium complexity, and both collapse at high complexity. Such results indicate unexpected scaling phenomena in reasoning learning and find that there is somewhere a boundary for the current reasoning model, regardless of computational cost.

**Questions:**

Could you provide the results for Qwen and QwQ on the puzzle problems?
I noticed that Qwen appears only in Figure 6 and is not included in earlier figures. Is there a particular reason for this omission?

Could you clarify how the algorithm is incorporated into the prompt?
It seems that providing the algorithm does not lead to significant performance differences in your results. However, as mentioned in [2], supplying correct exemplars can substantially improve performance on tasks like BlockWorld. Providing the algorithm should have similar effects. Could you share more details about your approach?



[1] Anil et al. Exploring Length Generalization in Large Language Models

[2] Zhao at al. Improving Large Language Model Planning with Action Sequence Similarity

**Ethical Concerns:**

["NO or VERY MINOR ethics concerns only"]

**Final Justification:**

Most of my concerns are well-discussed by the authors, but I do think the problem settings need to be improved, and its claim is a little bit too strong.

**Limitations:**

yes

**Quality:**

3

**Strengths And Weaknesses:**

strengths:

(1) Controllable Task Complexity:
The use of controllable puzzle environments enables systematic adjustment of problem complexity while preserving logical consistency. This allows for more principled experimental design compared to standard math benchmarks, which are often affected by data contamination and uncontrolled variability.

(2) Interesting and Timely Findings:
The discovery of the high-complexity collapse phenomenon in reasoning models is both interesting and relevant. The claim is carefully supported with experimental results, and the finding ties into current discussions on long-range modeling (LRM).



weaknesses:

(1) Lack of Theoretical Analysis on Complexity:
The paper does not clearly define task difficulty or provide theoretical assumptions about problem complexity. Different tasks may follow different scaling laws as their difficulty increases considering their parameters. Some problems like River Crossing can even suddenly becomes unsolvable when increasing its number of problem settings. Even though this is an empirical study, offering at least a provisional analytical model—such as relating compositional depth to compute scaling, or explaining reduced token usage as problems become harder—would strengthen the explanation of the observed collapse phenomenon.

(2) Confounding Variable: LLMs’ Long-Context Length Ability:
The collapse in model reasoning may be partially due to limitations in the model’s long-context understanding. Since the models were not trained on sequences of this length, they may lose track of earlier information. Exploring strategies such as dynamically summarizing previous history or breaking problems into step-by-step segments could help address these issues. Prior work [1] also suggests that performance collapse can occur when models hit output length limits or fail to generalize to longer contexts.

(3) Confounding Variable: Output Length Constraints:
LLMs sometimes terminate generation prematurely if the output is too long, which might contribute to the observed collapse in reasoning. This tendency to stop early should be considered as a potential explanation.

(4) Lower-than-Expected Blocksworld Performance:
In Figure 6 of [2], blocksworld performance is reported to be much higher than in this paper. More details on the experimental settings and possible reasons for this discrepancy would be helpful.

[1] Anil et al. Exploring Length Generalization in Large Language Models

[2] Zhao at al. Improving Large Language Model Planning with Action Sequence Similarity

---

> ### Author Rebuttal · Authors · 2025-07-30
>
> Thank you for dedicating your time and expertise to review our submission. Please find our responses below.
>
> > Output Length Constraints: LLMs sometimes terminate generation prematurely if the output is too long, which might contribute to the observed collapse in reasoning. This tendency to stop early should be considered as a potential explanation.
>
> Our empirical evidence demonstrates that collapse behavior observed in experiments occur within the output limits in all puzzles. For the Tower of Hanoi specifically, with its exponential growth raising questions of context-limit failures, reasoning models begin to collapse at N=7-8, corresponding to 127-255 moves (as shown in Figure 4 and 10 in appendix) which is well within the context limits. More importantly, if we look deeper into the failure cases (like in Figure 8c and 11 in appendix), we see that the first failure move actually happens much sooner than the final move. For example, for ToH with N=10 (requiring 1023 moves), failure typically occurs within the first $\sim$100 moves (10% of solution length); for N=8, which requires 255 moves, failure occurs around $\sim$40 moves (15% of solution length). This indicates that model failures and particularly collapse behavior happen much earlier and are not due to premature outputs because of generation limits. To account for context limit concerns for large values of N in ToH, we have excluded N>12 from experiments. We do plan to update the camera-ready accordingly.
> Notably, N>12 lies within the third region, post-collapse (N=8), so this modification does not affect our core findings for this puzzle.
>
> > The paper does not clearly define task difficulty or provide theoretical assumptions about problem complexity. ... Different tasks may follow different scaling laws as their difficulty increases considering their parameters.
>
> The theoretical scaling of computational complexity for puzzles in terms of compositional depth with respect to parameter N are already provided in the Appendix A.3 (e.g. Figure 9 shows the compositional depth with N for puzzles). You are correct that different tasks may follow different theoretical scaling with respect to N (as shown in Figure 9), however, theoretical computational complexity does not help here as it does not reflect the model behavior. LLMs (or LRMs) are complex artifacts and how they approach complexity does not necessarily correspond to the actual theoretical computational complexity of the problem and more to the learned solution distributions from their training data.
>
> > Even though this is an empirical study, offering at least a provisional analytical model—such as relating compositional depth to compute scaling, or explaining reduced token usage as problems become harder—would strengthen the explanation of the observed collapse phenomenon.
>
> Thank you for the helpful suggestion. We have already examined accuracy vs. compositional depth in Appendix A.3 (Figure 10). Following your suggestion, we also analyzed inference compute (tokens) vs. compositional depth and found very similar patterns to Figure 6: models increase compute up to a complexity point, then counterintuitively decrease it for higher compositional depths. We also observe inconsistent compute allocation across puzzles, with models sometimes spending more tokens on problems with lower compositional depth and vice versa—further supporting that their approach to complexity doesn't necessarily align with actual computational complexity. We will make sure to include this analysis in the camera ready.
>
> Regarding reduced reasoning effort on harder problems, we hypothesize this happens when harder problems fall outside the frequent training distribution, causing uncertainty and scaling limits in compute allocation. While we cannot verify this without access to training data, our empirical results support this speculation.
>
> > LLMs’ Long-Context Length Ability: The collapse in model reasoning may be partially due to limitations in the model’s long-context understanding.. strategies such as dynamically summarizing previous history or breaking problems into step-by-step segments could help.. Prior work [1] also suggests that performance collapse can occur when models hit output length limits or fail to generalize to longer contexts.
>
> Thank you for the suggestion. While we acknowledge that longer move sequences may reduce the chance of correct reasoning chains, our experiments and others' findings confirm that long context is not the primary cause for the observed collapse behavior in these puzzles. First of all, collapse also occurs in short-context tasks—e.g., River Crossing with N=3 (11 moves) and Checker Jumping with N=4 (24 moves)—indicating that failure is not solely due to sequence length.
> Second, our ablations using deterministic decoding ($\tau=0$) on open-source DeepSeek-R1 also show that removing sampling does not prevent collapse. In fact, sampling often delays it: eg ToH collapses at N=8 with $\tau=0$ but extends to N=9 with sampling (18.2% accuracy at N=8). In Blocks World also $\tau=0$ collapses at N=4 versus N=30 with sampling. These results suggest that sampling over long contexts is not the primary cause of collapse.
>
> Also, a recent paper (Varela et al., 2025) has explored stepwise prompting and agentic strategies on these tasks (e.g., ToH, River Crossing), breaking them to step-by-step segments as reviewer suggested, but still finds similar collapse behavior (eg $\sim$N=8 for ToH), supporting our conclusion that the collapse behavior lies in the model’s ability to generalize compositional reasoning with complexity—not just in handling long contexts.
>
>
> > In Figure 6 of [2], blocksworld performance is reported to be much higher than in this paper. More details on the experimental settings and possible reasons for this discrepancy would be helpful.
>
> Thank you for raising this question. Performance discrepency is because of different designs in initial and target block configurations. [2] is only testing on the non-reasoning models (e.g. gpt-4o, gemini 1.5 pro, llama-3.1). Our design is set to be more challenging for the recent reasoning models requiring more disassembly and reassembly of the stacks (eg. alternating between blocks from different stacks to reach target). Details of our design for initial and target block configurations are already provided in the Appendix A.1.4 L963-971. We will make sure to clarify this further in the updated version.
>
>
> > Could you clarify how the algorithm is incorporated into the prompt? It seems that providing the algorithm does not lead to significant performance differences in your results. However, as mentioned in [2], supplying correct exemplars can substantially improve performance on tasks like BlockWorld. Providing the algorithm should have similar effects. Could you share more details about your approach?
>
> In our experiments, we tested different algorithm incorporation methods: algorithm alone as scratchpad, and algorithm with execution examples on simple problems. Both settings showed very similar performance where algorithm provision does not change the observed collapse behavior. Details of these experiments are already provided in Appendix A.2 L1025.
>
> One may speculate that this observed performance for algorithm provision might be only true for ToH where algorithm is already more known for models; so in response, we also conducted additional experiments on Checker Jumping and our results show very similar behavior to what we observed on ToH where algrithm may lead to slight boost in the accuracy for some N but it generally does not change the collapse point. This behavior is also puzzling for us and we think it might be one of the key limitations of recent LRMs. Notably, [2] has only tested this with non-reasoning models, while our focus is on reasoning models. LRMs' failure in following algorithm steps might also partially relate to their weaker instruction-following capabilities (as shown by some recent works eg [3]). Exploring these puzzling behaviors and limitations in LRMs could be an important direction for future research.
>
> [3] Li et al., When Thinking Fails: The Pitfalls of Reasoning for Instruction-Following in LLMs, 2025
>
>
> > Could you provide the results for Qwen and QwQ on the puzzle problems? I noticed that Qwen appears only in Figure 6 and is not included in earlier figures. Is there a particular reason for this omission?
>
> Figure 6 includes R1-Distilled-Qwen specifically to compare with DeepSeek-R1 for examine of LRMs. Our main experiments which is the core for other figures focus on two reasoning/non-reasoning model pairs: DeepSeek-R1 vs V3, and Claude 3.7 Sonnet w. & w/o thinking.
> In response to your suggestion, we also conducted additional experiments during rebuttal on QwQ-32B reasoning and its corresponding non-reasoning pair Qwen-2.5-32B. Results show very similar collapse behavior to other model pairs and will be included in the camera-ready version.
>
> ---
> Thank you once again for the thoughtful review. We noticed that several concerns raised in the reviews echo some misunderstandings about our methodology and findings. In our response, we have addressed each point with detailed evidence and clarification, while incorporating revisions where appropriate. We hope that our rebuttal address the reviewer's concerns, and if so, they would consider updating their score. We’d be more than happy to engage in further discussions.

---

> > ### Comment · Reviewer_3cZ8 · 2025-08-04
> >
> > Thanks for your comprehensive rebuttal. I choose to keep my score.

---

> > > ### Author Response · Authors · 2025-08-04
> > > **Response from Authors**
> > >
> > > Dear Reviewer 3cZ8,
> > >
> > > Thank you for acknowledging our rebuttal. If there were any leftover concerns, we would sincerely appreciate the opportunity to clarify them—before the discussion period for the authors ends. We believe our responses have addressed in detail the full set of questions/concerns you raised.
> > >
> > > Should there be no additional concerns, we kindly ask you to consider revising your score.
> > >
> > > Thank you for your time and thoughtful feedback!

---

> > > > ### Author Response · Authors · 2025-08-05
> > > > **New Results from Authors**
> > > >
> > > > Dear Reviewer 3cZ8,
> > > >
> > > > Regarding the new experiments with QwQ-32B and Qwen-2.5-32B, we have new results based on completed experiments. While we cannot submit a new revision here and any link to updated figures due to conference policy, we provide the summary of results here, and the full experiments will be added to the camera-ready version (Figures 4,5,6, and 8 will be updated to include the new model)
> > > >
> > > > As shown in the table below, we observe very similar behavior for this new model pair compared to other pairs studied in the paper (slightly worse performance than DeepSeek-R1 vs V3). Specifically, we still observe the three regimes of complexity for this reasoning vs non-reasoning model pair. For QwQ-32B, collapse occurs similar or slightly earlier than for other LRMs (N=7 for Tower of Hanoi, N=3 for Checker Jumping, N=10 for Blocks World, and N=3 for River Crossing). Consistent with prior observations, QwQ-32B also shows a counterintuitive reduction in reasoning effort (thinking tokens) around and beyond the collapse point. Lastly, in algorithm-provision experiments on ToH and Checker Jumping, QwQ-32B behaves similarly to other LRMs: providing the explicit solving algorithm—requiring only step-by-step logical execution—does not shift the model behavior, and particularly the collapse point.
> > > >
> > > > | Puzzle         | QwQ-32B Collapse | Qwen-2.5-32B Collapse | Comparison to R1 & R1-Distilled-Qwen-32B            | QwQ Thinking Tokens (Reasoning Effort)             |
> > > > |----------------|------------------|-----------------------|-----------------------------------------------------|------------------------------|
> > > > | Tower of Hanoi | N = 7            | N = 5                 | Earlier than R1 (N=9), later than R1-Distilled-Qwen-32B (N=6)   | Starts to drop after N=5              |
> > > > | Checker Jumping | N = 3            | N = 2                 | Similar to both R1 and R1-Distilled-Qwen-32B (N=3)              | Starts to drop after N=2              |
> > > > | Blocks World | N = 10           | N = 3                 | Earlier than R1 (N=25–30), later than R1-Distilled-Qwen-32B (N=3)| Starts to drop after N=15 |
> > > > | River Crossing | N = 3            | N = 3                 | Similar to other models (N=3)                       | Starts to drop after N=3              |
> > > >
> > > > We hope that this help to address the reviewer's remaining concerns. We’d be happy to also engage in further discussions before the discussion period ends in one day.

---

### Note · Authors · 2025-08-15

Dear Area Chairs and Reviewers,

Thank you for dedicating your time and expertise to review our submission. We are grateful for the thorough and constructive review process. We are also glad that our rebuttals were able to resolve the concerns of most reviewers, with Reviewer K8Ta, P5aZ, and rPRW kindly raising their scores in light of the discussions.

We believe that we have addressed all the concerns raised by reviewer 3cZ8 in our rebuttal. Unfortunately, we did not have the opportunity to engage with the reviewer during the discussion to address any remaining concerns.

The primary concerns raised by Reviewer 3cZ8 centered on the following key areas:

**Lack of Theoretical Analysis on Complexity.** We provided empirical evidence that theoretical complexity scaling between problems does not necessarily align with observed reasoning model behavior (as discussed in Appendix A.3). Following the reviewer’s suggestion, we also added a new analysis of compute vs. compositional depth in appendix, which further supports these empirical evidence.

**Long-Context as Primary Cause of Failures.** We clarified that collapse also occurs in short-sequence tasks (e.g., River Crossing, Checker Jumping) and with new experiments with temperature 0 we showed that sampling over long context is not the primary cause for the observed collapse behavior.

**Output Limits as Primary Cause of Failures.** To account for context limit concerns for large values of N in ToH, we excluded N>12 from this environment. We also clarified with empirical evidence that collapse behavior always occur within the output limits with failure moves happenning much earlier and not due to premature outputs.

**Different Results on Blocksworld.** We clarified that discrepencies are due to different initial/target block configurations in our setup (Appendix A.1.4), which were designed to be more challenging to better stress recent reasoning models.

**Additional Model Backbone (QwQ & Qwen-2.5-32B).** In response to reviewer suggestion, we conducted new experiments for Qwen-based reasoning/non-reasoning model pair, and find very similar behavior consistent with other model pairs.

We hope this summary clarifies that all raised concerns were addressed in the rebuttal. We sincerely appreciate the time chairs and all reviewers have invested in the assessment of our submission and respectfully request the ACs to consider these points in the final decision.

Respectfully,

Authors of Submission #24872

---

### Decision · Program_Chairs · 2025-09-17

**Decision:**

Accept (poster)

**Comment:**

In ‘The illusion of thinking’ the authors consider the strengths and limits of reasoning models be testing them with four different problem classes where the model must find a sequence of legal moves to transform an initial state to a specified end state.  Problems in each class can be constructed to vary in what the authors call ‘complexity’, operationalized as a variable N that controls the minimum number of moves required to achieve a solution.  For example, in the tower of Hanoi problem, a stack of N discs must be moved from one peg to another under constrained rules.  As N increases, then number of moves required to solve the task increases.  The paper compares matched models with base and ‘thinking’ variants, and reports several fairly consistent patterns.  First, for low-complexity problems, the base and thinking variants perform well, but the thinking variants produce more tokens.  Second, for intermediate complexity problems, thinking models outperform base variants, but begin to fail while generating longer and longer chains of thought.  Third, there is a critical difficulty level (specific to each model and problem type) where the thinking models performance ‘collapses’, i.e. its accuracy falls to zero.  Beyond this point, rather than continuing to generate even longer trains of thought on even harder problems, the models actually end their responses after somewhat shorter sequences.  Another finding is that thinking models tend to generate correct answers within their thinking traces well before they stop ‘thinking’ – a phenomenon others have described (as the authors acknowledge) called ‘overthinking’.  The authors are able to assess whether particular solution steps are valid or not within thinking traces and find that for simpler problems, earlier solution steps can sometimes be more accurate, while for harder problems, earlier steps tend to be incorrect while later steps are more likely to be accurate.

The reviewers identified several strengths of the work, including the timeliness of understanding the performance of reasoning models, the use of problems that vary parametrically in the number of solution steps required, the value of comparing base and reasoning variants of the same model, and the complex three stage pattern that these comparisons contributed to delineating.  All of the reviewers cited weaknesses; some of these reflected misunderstanding or missed information on first reading, while others were resolved with clarifications after a productive back and forth exchange.  Still others were addressed with additional results.
All reviewers acknowledged the responsiveness of the authors to their concerns.  In the end, 3 of the 4 reviewers gave the paper a 4/borderline accept rating, with the fourth stuck with their initial 3/borderline reject rating.

After reading the paper and the exchange with the reviewers, I find myself in agreement with the majority opinion that the paper falls in the borderline acceptable range.  The stated plans for revision make me guardedly optimistic that the final product will be much clearer and stronger than the original, leading me to recommend acceptance of the article a poster, space permitting.

Whether the article is finally accepted or not, I hope to authors will pursue the work to publication.  Producing a strong publication will require great care in ensuring that many points are clearly addressed within the body of the article to avoid the confusions some of the reviewers experienced.   For example, ‘complexity’ is used quite vaguely in the main text – the discussion in Appendix 3 gives it considerably more meaning and this should be more fully addressed in the text.  Furthermore, the authors appear to vacillate on the extent to which it is reasonable to assume the models have not been extensively contaminated by related problems in their training data.  I’d suggest considerable caution here, and not just in the case of the tower of Hanoi problem.  The final version of the paper should also reconsider the authors’ characterization of the tendency reasoning models sometimes show to produce shorter reasoning traces when given problems beyond the complexity threshold where accuracy drops to 0 (e.g., in Fig 1, bottom middle, when the number of disks in the tower of Hanoi problem is 15 or 20). The authors characterize this (line 72) in terms of ‘reducing reasoning effort’.  However, this may not be the right way to characterize the phenomenon. Like one of the reviewers, I felt it was very important to know that these were not cases where the model simply gives up – instead they were cases where the model produced its final (always false) solution more quickly as complexity increased.  The fact that the model reaches incorrect solutions more quickly as complexity increases is consistent with the idea that the model may lose sensitivity to whether a tentative solution is valid as complexity becomes to great.

More generally, I think a great deal of humility is required in offering interpretations of the patterns the authors have uncovered.  I hope they with read the paper over with a very critical eye with this point in mind as they finalize the paper for publication.